# Meta-Learning Objectives for Preference Optimization

**Carlo Alfano**[*][†]
Department of Statistics
University of Oxford

**Silvia Sapora**[*]
Department of Statistics
University of Oxford

**Jakob N. Foerster**
Department of Engineering
University of Oxford

**Patrick Rebeschini**
Department of Statistics
University of Oxford

**Yee Whye Teh**
Department of Statistics
University of Oxford

## Abstract

Evaluating preference optimization (PO) algorithms on LLM alignment is a challenging task that presents prohibitive costs, noise, and several variables like model size and hyper-parameters. In this work, we show that it is possible to gain insights on the efficacy of PO algorithm on simpler benchmarks. We design a diagnostic suite of MuJoCo tasks and datasets, which we use to systematically evaluate PO algorithms, establishing a more controlled and cheaper benchmark. We then propose a novel family of PO algorithms based on mirror descent, which we call Mirror Preference Optimization (MPO). Through evolutionary strategies, we search this class to discover algorithms specialized to specific properties of preference datasets, such as mixed-quality or noisy data. We demonstrate that our discovered PO algorithms outperform all known algorithms in the targeted MuJoCo settings. Finally, based on the insights gained from our MuJoCo experiments, we design a PO algorithm that significantly outperform existing baselines in an LLM alignment task.

## 1 Introduction

Learning from human preferences (Christiano et al., 2017) is a paradigm which enables the alignment of machine learning systems to relative human preferences, without requiring access to absolute rewards. While the framework was developed for robotic and games applications with experiments on MuJoCo simulations and Atari (Akrour et al., 2012; Biyik & Sadigh, 2018; Ibarz et al., 2018), this paradigm has been successfully applied to Large Language Models (Team et al., 2023; Achiam et al., 2023). In particular, fine-tuning pre-trained LLMs with human preferences has become a popular strategy to adapt them to specific tasks and to improve their safety and helpfulness.

Within this framework, Reinforcement Learning from Human Feedback (RLHF) is one of the most popular methods. It consists in learning a reward function using a preference dataset and then optimizing the estimated reward using Reinforcement Learning methods such as Proximal Policy Optimization (PPO) (Schulman et al., 2017). However, the training pipeline of RLHF is quite complex, which is why implicit approaches such as Direct Preference Optimisation (DPO) (Schulman et al., 2017) have gained traction thanks to their simplicity. These methods do not learn a reward model but estimate it implicitly using the policy of the agent. Many follow ups to DPO have been proposed (Yuan et al., 2023; Zhao et al., 2023; Azar et al., 2024; Xu et al., 2024a; Hong et al., 2024; Park et al., 2024; Meng et al., 2024), but comparing their performance in LLM alignment is a complex task that incurs high costs, noise, and the inherent difficulty in judging a response better than another.

In this work, we provide a comprehensive analysis of PO algorithms, examining their behavior on automatically generated preference datasets. We return to the roots of RLHF by performing this

---

[*]Equal contribution, order decided by coin flip.

[†]Now at Amazon.

39th Conference on Neural Information Processing Systems (NeurIPS 2025).

analysis in MuJoCo environments and datasets, where the underlying ground-truth reward structure is well defined and offers a clear performance metric to compare agents. In particular, we design a task where a pre-trained agent has to adhere to a new stylistic constraint, emulating the typical conditions of LLM fine-tuning. Our findings indicate that many PO algorithms present distinct failure modes when applied to specific mixed-quality or noisy datasets.

Moreover, we introduce a framework for finding PO algorithms. Specifically, we define a class of PO algorithms based on mirror descent (Nemirovski & Yudin, 1983), which generalizes DPO and ORPO for particular choices of the mirror map. We then show that this class can be easily parametrized and searched using evolutionary strategies (ES), optimizing for the final performance of the trained policy, as measured by the ground truth reward.

For each setting we consider, we discover an algorithm that significantly outperforms all baselines. Analyzing the discovered algorithms, we find that the main difference between them and the baselines is that they keep optimizing the policy of the agent well after the probability of generating the chosen trajectory has surpassed the probability of generating the rejected one. We use this insight to design a new PO algorithm, Temporally-Aware Mirror Preference Optimization (TeMPO), which demonstrate promising results in an LLM alignment task. We summarize our contributions below.

1. We perform a systematic evaluation of eight existing PO algorithms on automatically generated preference datasets with varying levels of data quality, noise levels and initial policy. We see that most existing algorithms struggle when dealing with noise and mixed-quality data.

2. We introduce a novel family of offline PO algorithms using mirror descent, named Mirror Preference Optimization (MPO), which can be easily parameterized and explored via ES.

3. For both noisy and mixed-quality settings, we find and describe a PO algorithm within our framework that largely outperforms all the considered baselines in our MuJoCo benchmark.

4. We demonstrate that takeaways from our analysis on the MuJoCo setting, as well as the characteristics of the discovered PO algorithms, can be successfully transferred onto LLM tasks. In particular, we show that our TeMPO algorithm significantly improves upon the baselines.

## 2   Preliminaries

Let $\mathcal{M} = (\mathcal{S}, \mathcal{A}, P, r, T, \mu)$ denote an episodic Markov Decision Process, where $\mathcal{S}$ and $\mathcal{A}$ are respectively the state and action spaces, $P(s' \mid s, a)$ is the transition probability from state $s$ to $s'$ when taking action $a$, $r(s, a) \in [0, 1]$ is the reward function, $T$ is the maximum episode length, and $\mu$ is a starting state distribution. A policy $\pi \in (\Delta(\mathcal{A}))^{\mathcal{S}}$, where $\Delta(\mathcal{A})$ is the probability simplex over $\mathcal{A}$, represents the behavior of an agent on an MDP, whereby at state $s \in \mathcal{S}$ the agents takes actions according to the probability distribution $\pi(\cdot \mid s)$. Let $\tau = \{(s_t, a_t)\}_{t=0}^{T-1}$ denote a trajectory of length $T$ and, with a slight overload of notation, let $\pi(\tau) = \prod_{t=0}^{T-1} \pi(a_t \mid s_t)$ and $r(\tau) = \sum_{t=0}^{T-1} r(s_t, a_t)$. Lastly, let $\pi(\cdot \mid \tau)$ be a distribution over $(\Delta(\mathcal{A}))^T$ defined as $\pi(\cdot \mid s_0) \times \cdots \times \pi(\cdot \mid s_{N-1})$.

Our objective is to find a policy $\pi^\star$ that maximizes the expected cumulative reward of an episode, that is

$$\pi^\star \in \operatorname*{argmax}_{\pi} \mathbb{E}_{\tau \sim (\mu, \pi, P)} r(\tau) := \operatorname*{argmax}_{\pi} \mathbb{E}_{s_0 \sim \mu, a_t, s_{t+1}} \sum_{t=0}^{T-1} r(s_t, a_t), \tag{1}$$

where $a_t \sim \pi(\cdot \mid s_t)$ and $s_{t+1} \sim P(\cdot \mid s_t, a_t)$. Let $\mathcal{D} = \{(s_0^i, \tau_w^i, \tau_l^i)_{i=1}^N\}$ be a preference dataset, where each tuple $(s_0, \tau_w, \tau_l)$ consists of a starting state $s_0$ and two trajectories with starting state $s_0$. Each pair of trajectories is ranked by a judge, who determines a chosen trajectory $\tau_w$ ("win") and a rejected trajectory $\tau_l$ ("lose"), based on the cumulative rewards $r(\tau_w)$ and $r(\tau_l)$. Most settings assume the judge ranks trajectories according to the Bradley-Terry model (Bradley & Terry, 1952), whereby the probability of choosing $\tau_w$ over $\tau_l$ is defined as

$$\mathbb{P}(\tau_w \succ \tau_l) = \frac{\exp(r(\tau_w))}{\exp(r(\tau_w)) + \exp(r(\tau_l))} = \sigma(r(\tau_w) - r(\tau_l)), \tag{2}$$

where $\sigma$ is the sigmoid function. In this work, we consider an offline training setting, where the agent aim to solve the optimization problem in (1) but only has access to the the dataset $\mathcal{D}$ and cannot collect further data. We also assume the agent does not have access to either the transition probability $P$, the reward function $r$, or the MDP $\mathcal{M}$.

## 2.1 Alignment to preference feedback

There are several algorithms in the literature to optimize the objective in (1) using a preference dataset $\mathcal{D}$. We describe supervised fine-tuning (SFT), DPO and ORPO, as they are among the most popular and as many methods can be seen as a variation of one of these algorithms.

**SFT** SFT is an initial alignment phase, where the policy $\pi_0$ is trained to imitate high-quality demonstration data. The starting policy $\pi_0$ is updated to minimize the cross-entropy loss $\ell(\pi, (s_0, \tau_w, \tau_l)) = -\log(\pi(\tau_w))$. We call *reference policy* $\pi_{\text{ref}}$ the policy obtained at the end of this procedure.

**DPO** Direct Preference Optimization (DPO) consists in solving a maximum likelihood estimation problem and a policy optimization problem in a single step. The maximum likelihood estimation problem is the one to find an estimate of the true reward function that governs how the preferences are expressed, that is

$$\widehat{r} \in \operatorname*{argmax}_{r_\theta} \mathbb{E}_{(s_0, \tau_w, \tau_l) \sim \mathcal{D}} \sigma(r_\theta(\tau_w) - r_\theta(\tau_l)), \tag{3}$$

for a parametrized reward class $\{r_\theta : \theta \in \Theta\}$. The policy optimization problem is the one to maximize the expected reward, that is

$$\pi^\star \in \operatorname*{argmax}_{\pi} \mathbb{E}_{s_0 \sim \mathcal{D}, \tau \sim (\pi, P)} \left[ \sum_{t=0}^{T-1} \mathbb{E}_{a \sim \pi(\cdot|s_t)} \widehat{r}(s_t, a) - \beta D_{\text{KL}}(\pi(\cdot|\tau), \pi_{\text{ref}}(\cdot|\tau)) \right], \tag{4}$$

where $D_{\text{KL}}$ represents the KL-divergence and is introduced to prevent the policy from moving too far away from the dataset distribution.

DPO merges these two problems by using the agent itself to implicitly represent the reward model. It consists in optimizing the objective

$$\pi^\star \in \operatorname*{argmax}_{\pi} \mathbb{E}_{(s_0, \tau_w, \tau_l) \sim \mathcal{D}} \left[ \log \sigma \left( \beta \left( \log \frac{\pi(\tau_w)}{\pi_{\text{ref}}(\tau_w)} - \log \frac{\pi(\tau_l)}{\pi_{\text{ref}}(\tau_l)} \right) \right) \right], \tag{5}$$

which is obtained by plugging the theoretical solution of (4) in the maximum likelihood problem in (3). Refer to Appendix C for details. Thanks to its simplicity, DPO has been widely adopted to fine-tune LLMs (Yuan et al., 2024; Jiang et al., 2024).

A known issue of DPO is that it pushes probability mass away from the preference dataset and to unseen responses (Xu et al., 2024b), which can cause the final policy to deviate significantly from the reference policy, even when the reference policy aligns well with human preferences. To mitigate this risk, DPO is usually applied for a few epochs.

**ORPO** ORPO further simplifies the training pipeline and addresses the distribution shift issue present in DPO. It merges the SFT and DPO steps into one, optimizing the unified objective

$$\pi^\star \in \operatorname*{argmax}_{\pi} \mathbb{E}_{(s_0, \tau_w, \tau_l) \sim \mathcal{D}} \Big[ \underbrace{\log \pi(\tau_w)}_{\text{SFT}} + \lambda \underbrace{\log \sigma \left( \log \left( \text{odds}_\pi(\tau_w) \right) - \log \left( \text{odds}_\pi(\tau_l) \right) \right)}_{\text{preference optimization}} \Big] \tag{6}$$

where $\text{odds}_\pi(\tau) = \pi(\tau)/(1 - \pi(\tau))$. ORPO gets rid of the need for a reference model by adding an SFT term to the preference optimization objective function, and uses this term to prevent the optimized policy from moving too far away from the dataset distribution. Additionally, the SFT term prevents pushing probability mass away from the preference dataset, addressing the distribution shift issue present in DPO.

Research on preference optimization has been very active and many methods have been proposed. We present a summary of some among the most popular algorithms in Table 4 and a brief discussion in Appendix A. Beyond these implicit algorithms, there are several other methods that explicitly solve the maximum likelihood problem in (3) and use the learned reward model to optimize the objective in (4) with an RL algorithm. Overall, RLHF is a superior approach and the industry standard, but is more computationally expensive and complex to implement. For a detailed discussion and comparison between DPO-like methods and PPO, refer to Appendix H.

## 2.2 Mirror Maps

We review the concept of mirror map, which will be needed when describing our methodology. For a convex set $\mathcal{X} \subseteq \mathbb{R}^{|\mathcal{A}|}$, a *mirror map* $h : \mathcal{X} \to \mathbb{R}$ is defined as a strictly convex, continuously differentiable and essentially smooth function[*] function that satisfies $\nabla h(\mathcal{X}) = \mathbb{R}^{|\mathcal{A}|}$. Essentially, a mirror map is a function whose gradient allows bijective mapping between the primal space $\mathcal{X}$ and the dual space $\mathbb{R}^{|\mathcal{A}|}$. The specific class of mirror maps that we are going to use is the $\omega$-potential mirror map class, to which most mirror maps considered in the literature belong.

**Definition 2.1** ($\omega$-potential mirror map Krichene et al. (2015))**.** For $u \in (-\infty, +\infty]$, $\omega \leq 0$, an $\omega$-*potential* is defined as an increasing $C^1$-diffeomorphism $\phi : (-\infty, u) \to (\omega, +\infty)$ such that

$$\lim_{x \to -\infty} \phi(x) = \omega, \ \lim_{x \to u} \phi(x) = +\infty, \ \int_0^1 \phi^{-1}(x) dx \leq \infty.$$

For any $\omega$-potential $\phi$, the associated mirror map is $h_\phi(\pi(\cdot|s)) = \sum_{a \in \mathcal{A}} \int_1^{\pi(a|s)} \phi^{-1}(x) dx$. When $\phi(x) = e^{x-1}$ we recover the negative entropy mirror map, while we recover the $\ell_2$-norm when $\phi(x) = 2x$ (refer to Appendix F). Mirror maps in this class are simple to implement in practice, where $\mathcal{A}$ is often large, as they can be parametrized by a scalar function instead of a multi-dimensional one. Additionally, the same $\omega$-potential $\phi$ can be used to generate mirror maps for different action spaces, allowing the insights obtained for one action space to easily generalize to others. An $\omega$-potential mirror map $h_\phi$ induces a *Bregman divergence* (Bregman, 1967), which is defined as

$$\mathcal{D}_{h_\phi}(\pi(\cdot|s), \pi'(\cdot|s)) := h_\phi(\pi(\cdot|s)) - h_\phi(\pi'(\cdot|s)) - \langle \nabla h_\phi(\pi'(\cdot|s)), \pi(\cdot|s) - \pi'(\cdot|s) \rangle,$$

where $\mathcal{D}_{h_\phi}(\pi(\cdot|s), \pi'(\cdot|s)) \geq 0$ for all $x, y \in \mathcal{Y}$. When $\phi(x) = e^{x-1}$, $\mathcal{D}_{h_\phi}$ is equivalent to the KL-divergence, while we recover the Euclidean distance when $\phi(x) = 2x$ (refer to Appendix F). When the Bregman divergence is employed as a regularization term in optimization problems, tuning the mirror map allows us to control the geometry of the updates of the parameters to be optimized, determining when to take large or small updates based on the current value of the parameters.

## 2.3 Evolution Strategies

OpenAI-ES (Salimans et al., 2017) is a popular method to be able to optimize non-differentiable functions and it has been widely used to meta-learn objectives (Lu et al., 2022; Jackson et al., 2024), as it obtains an unbiased estimate of the gradient (unlike second order gradient methods). The gradient $\nabla_\zeta F(\zeta)$ is estimated using:

$$\mathbb{E}_{\epsilon \sim \mathcal{N}(0, I_d)} \left[ \frac{\epsilon}{2\sigma} (\widehat{F}(\zeta + \sigma\epsilon) - \widehat{F}(\zeta - \sigma\epsilon)) \right],$$

where $\mathcal{N}(0, I_d)$ is the multivariate normal distribution, $d$ is the number of parameters, $\widehat{F}$ is an estimate of $F$, and $\sigma > 0$ is a hyperparameter regulating the variance of the perturbations.

## 3 Mirror Preference Optimization

We introduce Mirror Preference Optimization (MPO), a new framework for preference optimization that generalizes DPO and ORPO. We start by replacing the KL-divergence penalty term in the objective in (4) with a Bregman divergence and aim to solve the problem

$$\pi^\star \in \underset{\pi}{\operatorname{argmax}} \, \mathbb{E}_{s_0 \sim \mathcal{D}, \tau \sim (\pi, P)} \left[ \sum_{t=0}^{T-1} \mathbb{E}_{a \sim \pi(\cdot|s_t)} r(s_t, a) - \beta D_h(\pi(\cdot|\tau), \pi_{\text{ref}}(\cdot|\tau)) \right], \tag{7}$$

where $D_h$ is the Bregman divergence induced by a mirror map $h$. This new objective allows us to enforce different types of regularization, which, as we show later in the paper, can be tailored to account for specific properties of the preference dataset. Following the same intuition used to obtain the DPO objective, we have the following result.

**Theorem 3.1.** *Let $h_\phi$ be a $0$-potential mirror map and $\pi^\star$ be a solution to the optimization problem in (7). If $\pi_{\text{ref}}(a|s) > 0$ for all $s \in \mathcal{S}, a \in \mathcal{A}$, we have that*

$$r(\tau) = \beta\phi^{-1}(\pi^\star(\tau)) - \beta\phi^{-1}(\pi_{\text{ref}}(\tau)) + c(s_0), \tag{8}$$

*where $c(s_0)$ is a normalization constant that depends only on $s_0$.*

---

[*]A function $h$ is *essentially smooth* if $\lim_{x \to \partial\mathcal{X}} \|\nabla h(x)\|_2 = +\infty$, where $\partial\mathcal{X}$ denotes the boundary of $\mathcal{X}$.

We provide a proof for Theorem 3.1 in Appendix C. The next step is to model the reward using a classification problem based on the reward difference rather than the maximum likelihood problem in (3), as suggested by Tang et al. (2024). That is, our aim is to solve the optimization problem

$$\widehat{r} \in \operatorname{argmax}_{r_\theta} \mathbb{E}_{(s_0, \tau_w, \tau_l) \sim \mathcal{D}} g(r_\theta(\tau_w) - r_\theta(\tau_l)), \tag{9}$$

where $g$ is an increasing function. We give further details on this interpretation of reward modeling in Appendix D. By plugging (8) in the optimization problem in (9), we obtain the objective:

$$\pi^\star \in \operatorname{argmax}_\pi \mathbb{E}_{\mathcal{D}} \left[ g\big(\beta(\phi^{-1}(\pi(\tau_w)) - \phi^{-1}(\pi_{\text{ref}}(\tau_w)) - \phi^{-1}(\pi(\tau_l)) + \phi^{-1}(\pi_{\text{ref}}(\tau_l)))\big) \right], \tag{10}$$

where $\mathbb{E}_{\mathcal{D}}$ is equivalent to $\mathbb{E}_{(s_0, \tau_w, \tau_l) \sim \mathcal{D}}$.

**Two-step MPO (2S-MPO)**  We can use the objective in (10) to define a class of two-step PO algorithms, which consist of a preliminary SFT phase to obtain the reference policy $\pi_{\text{ref}}$ and a PO phase which optimizes (10). When $\phi = e^x$, the (10) is equivalent to (5) and we recover DPO.

**One-step MPO (1S-MPO)**  By adding an SFT term to (10) and by setting the $\pi_{\text{ref}}$ be the uniform distribution, we obtain a class of one-step PO algorithms. These algorithms consists in a single phase, where we optimize the objective

$$\pi^\star \in \operatorname{argmax}_\pi \mathbb{E}_{(s_0, \tau_w, \tau_l) \sim \mathcal{D}} \left[ \psi(\pi(\tau_w)) + \lambda g\left(\phi^{-1}(\pi(\tau_w)) - \phi^{-1}(\pi(\tau_l))\right) \right], \tag{11}$$

where $\psi$ is an $\omega$-potential. The term $\phi^{-1}(\pi_{\text{ref}}(\tau_l)) - \phi^{-1}(\pi_{\text{ref}}(\tau_w))$ has canceled out due to $\pi_{\text{ref}}$ being uniform. We note that setting $\pi_{\text{ref}}$ to be the uniform distribution is equivalent to replacing the Bregman divergence penalty in (7) with the mirror map $h(\pi(\cdot|\tau))$, which enforces a form of entropy regularization. When $\psi(x) = \log(x)$ and $\phi^{-1}(x) = \log(x/(1-x))$, (11) recovers the ORPO objective in (6).

**Temporally-Aware MPO (TeMPO)**  Lastly we design a variation of MPO that gradually switches from SFT to PO. TeMPO consists of single-phase algorithms that optimize the objective

$$\pi^\star \in \operatorname{argmax}_\pi \mathbb{E}_{(s_0, \tau_w, \tau_l) \sim \mathcal{D}} \left[ (1 - \alpha(t))\psi(\pi(\tau_w)) + \alpha(t) g\left(\phi^{-1}(\pi(\tau_w)) - \phi^{-1}(\pi(\tau_l))\right) \right], \tag{12}$$

where $\alpha : [0, 1] \to [0, 1]$ is an increasing function of the percentage of training progress.

The objectives in (10), (11), and (12) allow us to implement a variety of preference optimization algorithms, while benefiting from a theoretical justification. In the following, we will show that it is possible to parametrize and optimize $g$, $\psi$, and $\phi^{-1}$ to obtain new algorithms that outperform baselines. In Appendix E, we discuss why we chose Bregman divergences rather than $f$-divergences.

## 3.1   Meta Learning PO objectives

To search the space of PO algorithms we have defined, we employ a neural network parametrization for $g$, $\psi$, and $\phi^{-1}$, which we optimize using evolutionary strategies (Salimans et al., 2017).

Similarly to Alfano et al. (2024), we parameterize $g$, $\psi$ and $\phi^{-1}$ as a one layer neural network with 126 hidden units and non-negative kernels, where the activation functions are equally split among:

$$x, \ (x)_+^2, \ x^3, \ (x)_+^{1/2}, \ (x)_+^{1/3}, \ \log((x)_+), \ e^x, \ \tanh(x), \ \log(\operatorname{clip}(x)/(1 - \operatorname{clip}(x))),$$

where $(x)_+ = \max(x, 0)$ and $\operatorname{clip}(x) = \max(\min(x, 1), 0)$. The non-negative kernels and the increasing activation functions guarantee the monotonicity of $g$, $\psi$, and $\phi^{-1}$, while the several different activation functions facilitate expressing complex functions. To ensure that we are able to recover the DPO and ORPO objectives, we add $a \log(x)$, $b \log(x)$ and $c \log(x/(1 - x))$ to the final outputs of $g$, $\psi$ and $\phi^{-1}$, respectively, where $a, b, c \geq 0$.

To search for the best $g$, $\psi$ and $\phi^{-1}$ within this class, we employ the OpenAI-ES strategy. Denote by $\zeta$ the parameters of $g$, $\psi$ and $\phi^{-1}$ and by $\pi^\zeta$ the final policy obtained optimizing the objective in (11) when using the parametrized $\psi$ and $\phi^{-1}$. Lastly, let $F(\zeta)$ be the expected cumulative reward of $\pi^\zeta$, i.e. $F(\zeta) = \mathbb{E}_{\tau \sim (\mu, \pi^\zeta, P)} r(\tau)$. We then use Adam (Kingma & Ba, 2015) to update the parameters $\zeta$ using the estimated gradient. In practice, to compute (19), we sample 128 values of $\epsilon$ to obtain 256 perturbed objective functions. We then train 256 agents with the perturbed objective functions on an a preference dataset. To measure the value of each agent, i.e. $\widehat{F}(\zeta')$ for all perturbed $\zeta'$, we sample 100 trajectories on the target environment for each agent, and take the average cumulative reward as estimate for the value of the agent. Refer to Appendix G for further discussion and details on the ES methodology.

We consider both the case where we fix $g = \log \sigma$ and learn $\psi$ and $\phi^{-1}$, and the case where we learn all three functions. We perform the evolution on both the two-step and one-step MPO classes.

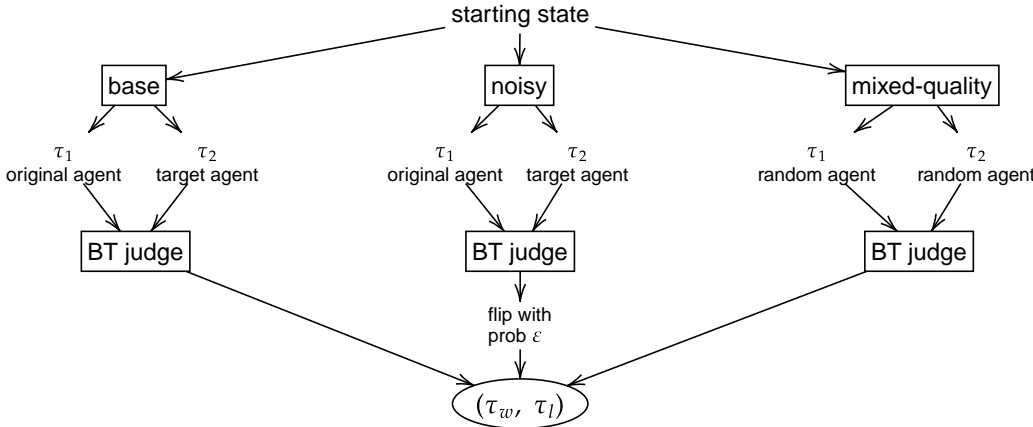

Figure 1: Data generation for MuJoCo experiments. For each pair of trajectories, which share the same starting state, two agents are chosen based on the data generating strategy. The trajectories are then judged by a Bradley-Terry Judge and, in the noisy setting, the labels are flipped with probability $\varepsilon$.

## 4 MuJoCo Experiments

Our first set of experiments is carried out on continuous RL tasks in MuJoCo. In particular, we show the performance of all the algorithms presented in Table 4 across several settings and we compare it with the performance of our discovered objectives. To maximize computational efficiency, all our MuJoCo experiments are implemented in JAX (Bradbury et al., 2018) using the brax (Freeman et al., 2021) and evosax (Lange, 2022) libraries. We provide an implementation of our methodology here and report hyper-parameters in Appendix K.

### 4.1 Tasks

To reproduce the typical conditions of LLM fine-tuning, which involve a pre-trained model, we consider a setting where the task is to adapt a pre-trained agent to meet the original objective while adhering to an additional stylistic constraint. Specifically, in the `Ant` environment, we start from an agent that has been pre-trained on the standard `Ant` goal of moving forward and enforce the objective of avoiding the use of one of its legs. This is accomplished by introducing the `Three-legged-ant` (TLA) environment, a modified version of `Ant` where utilizing the fourth leg results in significant penalties.

The offline preference optimization task is defined as follows. We train one agent (the original agent) in the original `Ant` environment, achieving a reward of 6000, and another (the target agent) in the `TLA` environment, which achieves a reward of 3900. For comparison, the original agent achieves a reward of 1700 in the `TLA` environment. We then generate a preference dataset of 1280 rows, each with two trajectories of length 1000 starting from the same state. Each trajectory is generated by either the original or the target agent, depending on the current setting. A Bradley-Terry judge ranks each pair of trajectories and declares a winner, based on their true cumulative reward. We consider three variations of the preference dataset, each meant to represent a common issue of real world data.

- **Base dataset**: for each pair of trajectories, one is generated by the original agent and one by the target agent.

- **Noisy dataset**: same as the base dataset but each chosen/rejected pair of labels given by the judge is flipped with probability $\varepsilon$.

- **Mixed-quality dataset**: each trajectory in the dataset is generated by an agent selected at random between the original and the target one. The resulting dataset will consist of, approximately, $25\%$ comparisons between two trajectories from the target agent, $50\%$ comparisons between trajectories of different agents, and $25\%$ comparisons between two trajectories of the original agent.

We also consider training a randomly initialized agent on the `Hopper` environment. This setting addresses the case where a behavior has to be learned from the preference dataset and there is no prior knowledge of the task available. We report the results of the experiments for this task in Appendix I.3.

Table 1: **Three Legged Ant (TLA)**. Performance of existing and discovered MPO algorithms on TLA, for various dataset settings. For each algorithm-dataset combination, we report the average value and standard error of 25 trained agents. For each discovered MPO algorithm, we specify on which setting it was discovered and report its performance across all settings (with fixed hyperparameters). We underline the highest (or two highest if their confidence interval overlaps) average performance among the human-designed algorithm and report in bold the overall highest, for each setting.

| | Base | Mixed Quality | Noisy ($\varepsilon = 0.1$) | Noisy ($\varepsilon = 0.3$) |
|---|---|---|---|---|
| RRHF (Yuan et al., 2023) | $2789 \pm 285$ | $2245 \pm 134$ | $1730 \pm 442$ | $330 \pm 552$ |
| SLiC-HF (Zhao et al., 2023) | $3255 \pm 66$ | $2478 \pm 54$ | $2329 \pm 289$ | $1135 \pm 224$ |
| DPO (Rafailov et al., 2024) | $3528 \pm 58$ | $2766 \pm 89$ | $3082 \pm 80$ | $1519 \pm 140$ |
| IPO (Azar et al., 2024) | $\underline{3618 \pm 44}$ | $2937 \pm 85$ | $\underline{3162 \pm 66}$ | $1133 \pm 115$ |
| CPO (Xu et al., 2024a) | $3450 \pm 55$ | $2322 \pm 208$ | $2967 \pm 58$ | $\underline{2000 \pm 35}$ |
| ORPO (Hong et al., 2024) | $3087 \pm 322$ | $2500 \pm 71$ | $2841 \pm 50$ | $1953 \pm 37$ |
| R-DPO (Park et al., 2024) | $2606 \pm 65$ | $2107 \pm 40$ | $2099 \pm 50$ | $1667 \pm 24$ |
| SimPO (Meng et al., 2024) | $\underline{3683 \pm 78}$ | $\underline{3117 \pm 185}$ | $2314 \pm 752$ | $-3828 \pm 341$ |
| SFT | $3287 \pm 62$ | $2344 \pm 40$ | $2733 \pm 45$ | $\underline{2049 \pm 33}$ |
| *Our algorithms ↓* | | | | |
| LPO | $3774 \pm 102$ | $2841 \pm 46$ | $3617 \pm 69$ | $1569 \pm 156$ |
| *With $g = \log \sigma$* | | | | |
| 1S-MPO (mixed-quality) | $3206 \pm 330$ | $3153 \pm 274$ | $1319 \pm 714$ | $-3967 \pm 382$ |
| 1S-MPO (noisy, $\varepsilon = 0.1$) | $3789 \pm 60$ | $3210 \pm 60$ | $\mathbf{3813 \pm 47}$ | $3279 \pm 83$ |
| 2S-MPO (mixed-quality) | $3595 \pm 57$ | $2785 \pm 78$ | $3197 \pm 58$ | $1687 \pm 58$ |
| 2S-MPO (noisy, $\varepsilon = 0.1$) | $3551 \pm 58$ | $2784 \pm 63$ | $3190 \pm 62$ | $1569 \pm 122$ |
| *With parametrized $g$* | | | | |
| 1S-MPO (mixed-quality) | $3560 \pm 333$ | $\mathbf{3627 \pm 79}$ | $3371 \pm 410$ | $2681 \pm 251$ |
| 2S-MPO (mixed-quality) | $3736 \pm 51$ | $3202 \pm 64$ | $3488 \pm 73$ | $2253 \pm 125$ |
| 1S-MPO (noisy, $\varepsilon = 0.1$) | $\mathbf{3861 \pm 79}$ | $3075 \pm 87$ | $3724 \pm 59$ | $1771 \pm 107$ |
| 2S-MPO (noisy, $\varepsilon = 0.1$) | $3701 \pm 52$ | $3178 \pm 59$ | $3490 \pm 95$ | $2074 \pm 136$ |
| 1S-MPO (noisy, $\varepsilon = 0.3$) | $\mathbf{3931 \pm 69}$ | $3244 \pm 55$ | $\mathbf{3834 \pm 82}$ | $\mathbf{3417 \pm 82}$ |
| *Temporally-aware* | | | | |
| TeMPO (1) | $3577 \pm 45$ | $2730 \pm 63$ | $3106 \pm 62$ | $1971 \pm 35$ |
| TeMPO (2) | $3625 \pm 49$ | $3088 \pm 69$ | $3443 \pm 53$ | $1988 \pm 39$ |
| TeMPO (3) | $3352 \pm 57$ | $2256 \pm 44$ | $2725 \pm 46$ | $1923 \pm 27$ |

## 4.2 Results

We provide the results of our experiments for TLA in Table 1, which reports the performance of several PO algorithm and of our discovered objectives. We performed a hyperparameter search for each algorithm-dataset combination and only report the performance of the best hyperparameters. All algorithms are run for 12 epochs over the preference dataset, with the exception of DPO, IPO, SimPO and R-DPO, which are run for 2 epochs after 10 epochs of SFT. We provide an additional noisy setting ($\varepsilon = 0.2$) and performance for other existing algorithms in Table 5 in Appendix I.1.

We notice that none of the human-designed algorithms manages to recover the performance of the target agent and that most of them experience a drop in performance in mixed-quality and noisy settings.

**Importance of SFT**   The first group within Table 1 shows that SFT plays a key role in the performance across different settings. SimPO, which does not contain an SFT term nor an SFT step, is at the top of the leaderboard on the base and the mixed-quality setting but performs poorly on all noisy settings. IPO and DPO, which do not contain an SFT term but have an SFT step, are among the top performers on the base, mixed-quality and low noise settings. Their performance finally drops when the noise level reaches 0.3. Lastly, the algorithms that present an SFT term in their objectives, e.g. CPO, ORPO and, obviously, SFT, exhibit a suboptimal performance in the base and mixed-quality settings but are much more robust to noise than the other algorithms.

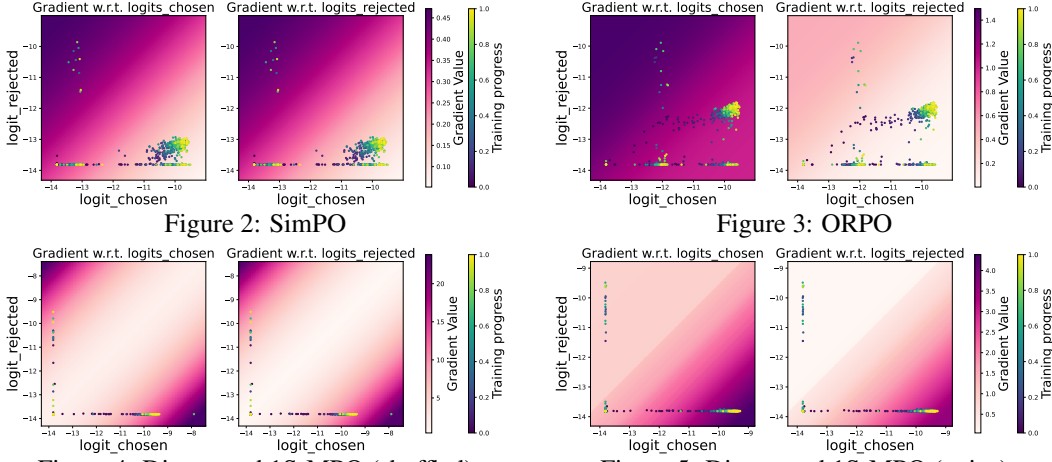

Figure 2: SimPO

Figure 3: ORPO

Figure 4: Discovered 1S-MPO (shuffled)

Figure 5: Discovered 1S-MPO (noisy)

Figure 6: Absolute value of the gradient of SimPO, ORPO, 1S-MPO (shuffled), and 1S-MPO (noisy, $\epsilon = 0.3$). The dots are sampled datapoints from the training distribution.

**Keep optimizing** Furthermore, while RRHF and SliC-HF are very similar, SliC-HF allows the PO part of the objective to be clipped when $\pi(\tau_w) > \pi(\tau_l) + \delta$, rather than when $\pi(\tau_w) > \pi(\tau_l)$. This modification leads to a higher performance on all tasks, demonstrating that it is beneficial to keep optimizing the policy even if $\pi(\tau_w) > \pi(\tau_l)$. To further stress this point, we consider the objective

$$\pi^\star \in \mathrm{argmax}_\pi \mathbb{E}_{(s_0,\tau_w,\tau_l)\sim\mathcal{D}}\big[\lambda \log \pi_\theta(\tau_w|x) - \log \pi_\theta(\tau_l|x)\big],$$

which we call Linear Preference Optimization (LPO). LPO corresponds to SliC-HF with $\delta = +\infty$ and obtains better results than most of the existing algorithms in Table 1, confirming that it is important to design objectives that do not flatten when $\pi(\tau_w) > \pi(\tau_l)$.

**Discovered objectives** Table 1 also reports the performance of our discovered objectives, for both the case where we set the monotonic transformation $g$ to be the logarithmic function and where we parametrize and learn it. We learn a separate objective for each dataset setting and report the performance of each learned objective on all settings. Differently from the human-designed algorithms, the discovered objectives recover the performance of the target agent in multiple instances and are more robust to the mixed-quality and noisy settings. We note that allowing the evolution procedure to learn $g$ leads to a better performance in all dataset settings. Additionally, we have that the objectives discovered within the two-step MPO class always have a lower performance than those within the one-step MPO class. This is probably due to the ability to modify the SFT term in the one-step MPO class, which is not present in the two-step class.

When exploring the one-step MPO class with $g = \log \sigma$, our discovery procedure always recovers a variation of CPO. That is, we obtain an objective that can be approximated as

$$\pi^\star \in \mathrm{argmax}_\pi \mathbb{E}_{(s_0,\tau_w,\tau_l)\sim\mathcal{D}}\big[\alpha \log(\pi(\tau_w)) + \lambda \log \sigma\big(\beta \log(\pi(\tau_w)) - \beta \log(\pi(\tau_l))\big)\big],$$

where the coefficients $\alpha$ and $\beta$ depend on the setting. In particular, we have a low value for $\alpha$ and a high value for $\beta$ in the noisy settings, while we observe the opposite in the mixed-quality setting. These results confirm the observations made on the hand-crafted objectives, whereby objectives with an SFT term are more robust to noise and objectives without are more robust to mixed-quality trajectories. When we search the two-step MPO class with $g = \log \sigma$, we recover DPO.

Figure 6 provides a visualization of the gradient of some of the objectives discovered when we parametrize and meta-learn $g$. In particular, we show the objectives discovered within the one-step MPO class on the mixed-quality and noisy ($\varepsilon = 0.3$) settings. For comparison, we provide the same plots for the ORPO and SimPO objectives. The hand-crafted algorithms present a larger gradient when $\pi(\tau_w) < \pi(\tau_l)$ and a smaller one when $\pi(\tau_w) > \pi(\tau_l)$, that is, they induce large updates when the data-point contradicts the current behaviour of the agent, and small otherwise.

The objective discovered on the noisy dataset has the opposite behavior, meaning that it only optimizes the more robust-to-noise SFT term when $\pi(\tau_w) < \pi(\tau_l)$. As $\log(\pi(\tau_w)) - \log(\pi(\tau_l))$

Table 2: **AlpacaEval LLM results**. We report win-rates and standard error (length controlled win-rates and standard error in parenthesis) for three combinations of base model and preference dataset. We report in bold font the highest winrate (length-controlled winrate) for each column.

| | gemma-7b[†], dpo-mix-7k[‡] | gemma-7b, capybara-7k[§] | mistral-7b[¶], dpo-mix-7k |
|---|---|---|---|
| CPO | $28.9\pm1.6$ $(21.9\pm0.2)$ | $29.2\pm1.6$ $(25.3\pm0.3)$ | $31.0\pm1.6$ $(21.7\pm0.3)$ |
| ORPO | $27.4\pm1.6$ $(21.5\pm0.2)$ | $28.2\pm1.6$ $(24.2\pm0.3)$ | $28.0\pm1.6$ $(21.1\pm0.3)$ |
| DPO | $30.7\pm1.6$ $(\mathbf{31.0\pm0.3})$ | $37.9\pm1.7$ $(32.8\pm0.2)$ | $32.8\pm1.6$ $(29.5\pm0.3)$ |
| SimPO | $32.5\pm1.7$ $(25.8\pm0.2)$ | $27.3\pm1.6$ $(23.6\pm0.2)$ | $28.1\pm1.6$ $(22.6\pm0.3)$ |
| TeMPO (1) | $31.5\pm1.6$ $(25.0\pm0.2)$ | $34.2\pm1.7$ $(30.0\pm0.2)$ | $39.6\pm1.7$ $(30.4\pm0.2)$ |
| TeMPO (2) | $27.8\pm1.6$ $(23.2\pm0.3)$ | $33.4\pm1.7$ $(29.6\pm0.2)$ | $36.6\pm1.7$ $(27.7\pm0.2)$ |
| TeMPO (3) | $\mathbf{35.4\pm1.7}$ $(29.1\pm0.1)$ | $\mathbf{39.4\pm1.7}$ $(\mathbf{33.8\pm0.1})$ | $\mathbf{41.6\pm1.7}$ $(\mathbf{30.6\pm0.2})$ |

becomes larger, it shifts toward increasingly large updates thanks to the PO term. A similar pattern appears in the mixed-quality dataset, where the objective also increases its updates as the difference $\log(\pi(\tau_w)) - \log(\pi(\tau_l))$ grows. The key distinction between these two losses is that the shuffled loss triggers high-gradient updates even when $\pi(\tau_w) < \pi(\tau_l)$. We highlight once more that both discovered objectives advocate to keep optimizing the policy even when $\pi(\tau_w) > \pi(\tau_l)$.

### 4.3 Including temporal awareness

We build an algorithm within the TeMPO family using the insights gained in the previous section. In particular, we keep the standard SFT loss, which was rediscovered in all our experiments, and use SimPO for the PO component of TeMPO, given its high performance and its ability to continue optimizing the policy even when $\pi(\tau_w) > \pi(\tau_l)$. That is, we define the objective

$$\pi^\star \in \arg\max_\pi \mathbb{E}_\mathcal{D}\big[(1-\alpha(t))\log\pi_\theta(\tau_w|x) + \alpha(t)\log\sigma(\beta(\log\pi_\theta(\tau_w|x) - \log\pi_\theta(\tau_l|x)) - \gamma)\big]. \quad (13)$$

We try three expressions for $\alpha$, designed to put most of the weight on the SFT component of (13) at the start of training and switch to PO towards the end of training:

$$(1): \ \alpha(t) = t; \quad (2): \ \alpha(t) = t^2; \quad (3): \ \alpha(t) = \sigma(20(x - 0.75)).$$

Table 1 shows promising results for temporally-aware PO algorithms, as the objective in (13) with the version (2) of $\alpha$ matches or surpasses all human-designed algorithms in all settings.

## 5 Experiments: LLM transfer

We show that the insights obtained in the MuJoCo environments can be transferred to the LLM alignment setting. In particular, we test the TeMPO algorithm in (13), which is designed to continue to optimize the policy even when $\pi(\tau_w) > \pi(\tau_l)$, as all our discovered algorithms do. We evaluate TeMPO on LLM alignment, comparing its performance against baselines for three combinations of base model and preference dataset, as shown in Table 2.

To tune the LLMs, we modify the Alignment Handbook library (Tunstall et al.) to include the TeMPO objective in (13). We evaluate the tuned LLMs against GPT-4, using the AlpacaEval library (Li et al., 2023) and `Llama-3.1-70B-Instruct` as a judge. For all combinations of starting LLM, dataset, and PO algorithm, we perform 4 update epochs and set the learning rate to 5e-5 and $\beta$ to 0.05. In the case of DPO and SimPO, we performed 3 epochs of SFT, with learning rate 5e-5, and 1 epoch of DPO/SimPO, with learning rate 5e-7 and $\beta = 0.05$. Refer to Appendix L for further details.

Table 2 shows the effectiveness of the TeMPO objective in (13), which presents a high winrate for all schedules of $\alpha$. In particular, TeMPO with the sigmoid schedule for $\alpha$ has the highest winrate in all tasks and the highest length controlled winrate in two out of three tasks. Additionally, TeMPO requires only one stage of training, while DPO and SimPO require two, i.e. SFT and PO.

---

[†]`https://huggingface.co/google/gemma-7b`

[‡]`https://huggingface.co/datasets/argilla/dpo-mix-7k`

[§]`https://huggingface.co/datasets/argilla/distilabel-capybara-dpo-7k-binarized`

[¶]`https://huggingface.co/mistralai/Mistral-7B-v0.3`

Table 3: We report win-rates and standard error on AlpacaEval (length controlled win-rates and standard error in parenthesis) for gemma-7b and two preference dataset.

| Algorithm | $\gamma$ | dpo-mix-7k | capybara-7k |
|---|---|---|---|
| SimPO | 0 | 30.50 (26.06) | 27.52 (25.12) |
| SimPO | 1 | 30.62 (25.35) | 27.52 (25.54) |
| SimPO | 2 | 32.50 (25.80) | 27.95 (25.71) |
| SimPO | 5 | 33.42 (28.22) | 27.02 (25.07) |
| SimPO | 10 | 33.60 (27.12) | 27.39 (25.00) |
| SFT+SimPO | 1 | 30.31 (26.55) | 35.07 (28.25) |

We also tested the static algorithms discovered in the MuJoCo environment. Those with $g = \log \sigma$, which rediscovered CPO, exhibited performance similar to CPO itself. In contrast, algorithms with a parameterized $g$ function, which learned to greedily optimize the policy even when $\pi(\tau_w) > \pi(\tau_l)$, performed poorly. We hypothesize that these algorithms overfit to the TLA task, where our datasets sufficiently cover the state-action space. On the other hand, in LLM tuning—where data is sparse relative to the environment—greedy methods are more susceptible to over-optimization. At the same time, the objectives discovered on TLA are only used to logits within -14 and -8, as shown in Figure 6, and have less regular shape outside of this subset.

In Table 3, we report additional results that allow us to better understand the influence of different components of TeMPO. In particular, we have tested SimPO with $\gamma = 0$, SimPO with $\gamma > 0$, and SFT + SimPO with $\gamma > 0$, where the last one consists of a one-step objective made of the addition between the SFT and the SimPO loss. Table 3 shows that having a large $\gamma$, which encourages large updates even when $\pi(\tau_w) > \pi(\tau_l)$, or adding an SFT term to the PO loss are both helpful in improving performance. The performance of TeMPO reported in Table 2 is still the highest, which means that temporal awareness also contributes to a better performance. We can therefore confirm that each of the insights we gained in the MuJoCo benchmark transfers to the LLM alignment setting.

## 6 Conclusion

We have introduced a novel framework for Preference Optimization algorithms, as well as a methodology for the automatic discovery of PO algorithms using evolutionary strategies. Through a systematic evaluation across diverse settings in MuJoCo environments, we have demonstrated that the performance of our discovered objectives consistently exceeds the performance of existing methods, particularly in noisy and mixed-quality datasets where many baselines underperform. Our analysis in MuJoCo also revealed a common shortcoming among current baselines: truncating the loss whenever $\pi(\tau_w) > \pi(\tau_l)$. Using this insight, we proposed a temporally-aware algorithm, TeMPO, that avoids such loss truncation and gradually switches from the SFT step to the PO step. We then tested this objective on an LLM fine-tuning task, achieving significant improvements over existing methods, thereby confirming the broader applicability of our approach.

### Acknowledgments and Disclosure of Funding

Carlo Alfano and Silvia Sapora are supported by the Engineering and Physical Sciences Research Council EP/W524311/. Jakob Foerster is partially funded by the UKI grant EP/Y028481/1 (originally selected for funding by the ERC). Jakob Foerster is also supported by the JPMC Research Award and the Amazon Research Award. Patrick Rebeschini is funded by UK Research and Innovation (UKRI) under the UK government's Horizon Europe funding guarantee [grant number EP/Y028333/1]. Yee Whye Teh acknowledges support from the Ministry of Digital Development and Information (MDDI) under the Singapore Global AI Visiting Professorship Program (Award No. AIVP-2024-002).

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

# A  Baseline PO Algorithms

Table 4: Overview of popular PO algorithms. The objective is to be maximized and $(\tau_w, \tau_l) \sim \mathcal{D}$.

| Method | Objective |
|--------|-----------|
| RRHF (Yuan et al., 2023) | $\lambda \log \pi_\theta(\tau_w|x) - \max\left(0, -\frac{1}{|\tau_w|} \log \pi_\theta(\tau_w|x) + \frac{1}{|\tau_l|} \log \pi_\theta(\tau_l|x)\right)$ |
| SLiC-HF (Zhao et al., 2023) | $\lambda \log \pi_\theta(\tau_w|x) - \max(0, \delta - \log \pi_\theta(\tau_w|x) + \log \pi_\theta(\tau_l|x))$ |
| DPO (Rafailov et al., 2024) | $\log \sigma \left(\beta \log \frac{\pi_\theta(\tau_w|x)}{\pi_{\text{ref}}(\tau_w|x)} - \beta \log \frac{\pi_\theta(\tau_l|x)}{\pi_{\text{ref}}(\tau_l|x)}\right)$ |
| IPO (Azar et al., 2024) | $-\left(\log \frac{\pi_\theta(\tau_w|x)}{\pi_{\text{ref}}(\tau_w|x)} - \log \frac{\pi_\theta(\tau_l|x)}{\pi_{\text{ref}}(\tau_l|x)} - \frac{1}{2\tau}\right)^2$ |
| CPO (Xu et al., 2024a) | $\log \pi_\theta(\tau_w|x) + \log \sigma(\beta \log \pi_\theta(\tau_w|x) - \beta \log \pi_\theta(\tau_l|x))$ |
| ORPO (Hong et al., 2024) | $\log \pi(\tau_w) + \lambda \log \sigma \left(\log\left(\text{odds}_\pi(\tau_w)\right) - \log\left(\text{odds}_\pi(\tau_l)\right)\right)$ |
| R-DPO (Park et al., 2024) | $\log \sigma \left(\beta \log \frac{\pi_\theta(\tau_w|x)}{\pi_{\text{ref}}(\tau_w|x)} - \beta \log \frac{\pi_\theta(\tau_l|x)}{\pi_{\text{ref}}(\tau_l|x)} + (\alpha|\tau_w| - \alpha|\tau_l|)\right)$ |
| SimPO (Meng et al., 2024) | $\log \sigma \left(\frac{\beta}{|\tau_w|} \log \pi_\theta(\tau_w|x) - \frac{\beta}{|\tau_l|} \log \pi_\theta(\tau_l|x) - \gamma\right)$ |

These methods include RRHF, which uses length-normalized log-likelihood, and SLiC-HF, which uses direct log-likelihood and incorporates SFT. The comparison also includes IPO, a theoretically-based approach that handles pairwise preferences differently than DPO. Another method, CPO, combines sequence likelihood as a reward with an SFT objective. Finally, R-DPO modifies the original DPO by adding regularization to prevent length exploitation.

# B  Related Work

**Automatic Discovery of Preference Optimization Loss Functions**  Several works in the literature have shown that it is possible to discover machine learning algorithms that outperform algorithms manually designed by researchers (Oh et al., 2020; Lu et al., 2022; Jackson et al., 2024; Alfano et al., 2024). An approach particularly relevant to our method is DiscoPOP by Lu et al. (2024), which leverages an LLM to discover objective functions for LLM tuning. They consider a different space of objective functions from us, as they replace the log-sigmoid in (5) with a generic loss function, following the framework built by Tang et al. (2024). Additionally, instead of searching over a space of parametrized functions, they ask the LLM to generate loss functions in code space. This distinction suggests that our approaches could be complementary, as the model discovered by DiscoPOP could be paired with our learned mirror map. Lastly, DiscoPOP optimizes its objective function directly on the final task, whereas we adopt a two-stage process—optimizing the loss function on a separate task (MuJoCo) and later transferring it to the LLM setting. This transferability underscores the broader applicability of our approach.

**Generalisations of DPO**  A generalization of DPO alternative to ours is $f$-DPO (Wang et al., 2023), which consists in replacing the KL-divergence in (1) with an $f$-divergence and then apply the same heuristic as DPO to obtain the final objective function. We note that the KL-divergence is the only $f$-divergence to be also a Bregman divergence, and vice-versa. They empirically demonstrate that different $f$-divergences lead to different balances between alignment performance and generation diversity, highlighting the trade-offs inherent to this class of algorithms. Huang et al. (2024) further explore this class of PO algorithm and individuate an $f$-divergence for which $f$-DPO is robust to overoptimization.

# C  Proof of Theorem 3.1

We provide here a proof for our main result, i.e. Theorem 3.1. The proof to obtain the DPO objective in (5) follows by taking $\phi = e^x$.

**Theorem C.1** (Theorem 3.1). *Let $h_\phi$ be a 0-potential mirror map and $\pi^\star$ be a solution to the optimization problem in (7). If $\pi_{\text{ref}}(a|s) > 0$ for all $s \in \mathcal{S}, a \in \mathcal{A}$, we have that*

$$r(\tau) = \phi^{-1}(\pi^\star(\tau)) - \phi^{-1}(\pi_{\text{ref}}(\tau)) + c(s_0), \tag{14}$$

*for all trajectories $\tau$, where $c(s_0)$ is a normalization constant that depends only on $s_0$.*

*Proof.* We use the KKT conditions to solve (7), i.e.

$$\pi^\star \in \underset{\pi}{\operatorname{argmax}} \, \mathbb{E}_{s_0 \sim \mathcal{D}, \tau \sim (\pi, P)} \left[ \sum_{t=0}^{T-1} \mathbb{E}_{a \sim \pi(\cdot|s_t)}[r(s_t, a)] - \beta D_h(\pi(\cdot|\tau), \pi_{\text{ref}}(\cdot|\tau)) \right]$$

We use the stationarity condition to obtain the equation

$$\nabla_{\pi(\tau)} \left[ \sum_{t=0}^{T-1} \mathbb{E}_{a \sim \pi(\cdot|s_t)}[r(s_t, a)] - \beta D_h(\pi(\cdot|\tau), \pi_{\text{ref}}(\cdot|\tau)) - \lambda \sum_{\tau':s_0 \in \tau'} \pi(\tau') - \lambda + \sum_{\tau':s_0 \in \tau'} \alpha(\tau')\pi(\tau') \right]$$

$$= r(\tau) - \beta\phi^{-1}(\pi(\tau)) + \beta\phi^{-1}(\pi_{\text{ref}}(\tau)) - \lambda + \alpha(\tau) = 0,$$

for all initial states $s_0 \in \mathcal{S}$ and for all trajectories $\tau$ starting from $s_0$. Rearranging, we obtain that

$$\pi(\tau) = \phi\big((r(\tau) + \beta\phi^{-1}(\pi_{\text{ref}}(\tau)) - \lambda + \alpha(\tau))/\beta\big).$$

Since $0 \notin \operatorname{dom} \phi^{-1}$, due to the definition of a 0-potential, and $\pi_{\text{ref}}(\tau) > 0$, we have that $\pi(\tau) > 0$ for all trajectories $\tau$. Invoking the complementary slackness condition, whereby $\alpha(\tau)\pi(\tau) = 0$ for all trajectories $\tau$, we have that $\alpha(\tau) = 0$ for all trajectories $\tau$. Therefore, we have that

$$r(\tau) - \beta\phi^{-1}(\pi(\tau)) + \beta\phi^{-1}(\pi_{\text{ref}}(\tau)) - \lambda = 0$$

The theorem statement is obtained by rearranging the last equation and denoting $c(s_0) = \lambda$ $\qquad\square$

## D  Reward Modeling

In Equation (9), we utilize the interpretation of reward modeling as a binary classification problem given by Tang et al. (2024), which we summarize here. Let $z = (\tau_1, \tau_2)$ be a pair of trajectories and $\ell \in \{-1, 1\}$ be the associated label that states whether $\tau_1$ is preferred to $\tau_2$ ($\ell = 1$) or not ($\ell = -1$). We want to find a function $\hat{\ell}(z) \in \mathbb{R}$ such that $\operatorname{sign}(\hat{\ell}(z))$ is a good estimate of $\ell$. For a dataset $\mathcal{D} = \{z_i, \ell_i\}_{i=1}^N$, the classification loss (or 0-1 loss) is

$$L(\widehat{\ell}, \mathcal{D}) = \mathbb{E}_\mathcal{D} \left[ 1 - \operatorname{sign}\left(\hat{\ell}(z) \cdot \ell\right) \right], \tag{15}$$

which is often approximated with a surrogate

$$L_f(\widehat{\ell}, \mathcal{D}) = \mathbb{E}_\mathcal{D} \left[ f\left(\hat{\ell}(z) \cdot \ell\right) \right], \tag{16}$$

for a function $f : \mathbb{R} \to \mathbb{R}$. This approximation is possible because, when $f$ is decreasing (or convex), (15) and (16) have the same minimizer, as we prove in the following. Denote $p_1(z) = \mathbb{P}(\ell = 1|z)$, then the conditional surrogate loss at $z$ is

$$L_f(\widehat{\ell}, x) = p_1(z)f\left(\hat{\ell}(z)\right) + (1 - p_1(z))f\left(-\hat{\ell}(z)\right). \tag{17}$$

The minimizer $\widehat{\ell^\star}$ of (16) is such that $\widehat{\ell^\star}(z)$ minimizes (17). While the minimizer to (17) might not be computable explicitly, we can show $\widehat{\ell^\star}(z) > 0 \iff p_1(z) > 1/2$ when $f$ is a decreasing function, meaning that $\widehat{\ell^\star}$ is also the minimizer of (15). Firstly, we have that

$$L_f(\widehat{\ell}, x) - L_f(-\widehat{\ell}, x) = p_1(z)f\left(\hat{\ell}(z)\right) + (1 - p_1(z))f\left(-\hat{\ell}(z)\right)$$

$$- p_1(z)f\left(-\hat{\ell}(z)\right) - (1 - p_1(z))f\left(\hat{\ell}(z)\right) \tag{18}$$

$$= (2p_1(z) - 1)\left(f\left(\hat{\ell}(z)\right) - f\left(-\hat{\ell}(z)\right)\right)$$

Since $f$ is decreasing, we have for all $\widehat{\ell}(z) > 0$ that $f(\widehat{\ell}(z)) < f(-\widehat{\ell}(z))$. Plugging this into (18), we obtain that, if $p_1(z) > 1/2$, $L_f(\widehat{\ell}, x) < L_f(-\widehat{\ell}, x)$ for all $\widehat{\ell}(z) > 0$. Therefore, the minimizer $\widehat{\ell^\star}(z)$ of (17) must be positive. The opposite can be proved in the same manner.

Equation (9) can be obtained by setting $f = -g$, for an increasing function $g$, and

$$\widehat{\ell}(z) = \widehat{r}(\tau_1) - \widehat{r}(\tau_2).$$

## E Bregman divergences vs $f$-divergences

In this section, we discuss why we use Bregman divergences rather than $f$-divergences in (7). Firstly, we note that, if the reference policy $\pi_{\text{ref}}$ is set to be the uniform distribution, Bregman divergences and $f$-divergences generate equivalent families of algorithms. Let $\phi$ be a 0-potential. Using a reasoning similar to that of Theorem 3.1, we have the following cases:

**Bregman divergence** $\quad r(\tau_w) - r(\tau_l) = \phi^{-1}(\pi^*(\tau_w)) - \phi^{-1}(\pi_{\text{ref}}(\tau_w)) - \phi^{-1}(\pi^*(\tau_l)) + \phi^{-1}(\pi_{\text{ref}}(\tau_l)),$

$\quad$ +uniform $\pi_{\text{ref}} \quad r(\tau_w) - r(\tau_l) = \phi^{-1}(\pi^*(\tau_w)) - \phi^{-1}(\pi^*(\tau_l)),$

$\quad$ $f$-**divergence** $\quad r(\tau_w) - r(\tau_l) = \phi^{-1}(\pi^*(\tau_w)/\pi_{\text{ref}}(\tau_w)) - \phi^{-1}(\pi^*(\tau_l)/\pi_{\text{ref}}(\tau_l)),$

$\quad$ +uniform $\pi_{\text{ref}} \quad r(\tau_w) - r(\tau_l) = \phi^{-1}(|\mathcal{T}|(\pi^*(\tau_w))) - \phi^{-1}(|\mathcal{T}|(\pi^*(\tau_l)))$
$$= \tilde{\phi}^{-1}(\pi^*(\tau_w)) - \tilde{\phi}^{-1}(\pi^*(\tau_l)),$$

where $\mathcal{T}$ is the set of all trajectories and $\tilde{\phi}^{-1}(x) = \phi^{-1}(|\mathcal{T}|x)$. The two resulting expressions are equivalent, meaning that our 1S-MPO class also includes the case of $f$-divergences. Our experiment on the 1S-MPO class therefore explore both Bregman divergences and $f$-divergences with ES and Table 1 reports the performance of the best divergence found within both divergence classes.

If the reference policy is not uniform, then the two algorithmic classes have the KL divergence as the only intersection. In this setting, we have chosen to focus on Bregman divergences as Wang et al. (2023) have already shown that, among the $f$-divergences they considered, the KL-divergence typically offers superior alignment performance. Since we wanted to explore a class of algorithms with the objective of finding better alternatives to the KL-divergence, we decided to consider a different generalization of the KL-divergence, i.e. Bregman divergences. In our experiments with $g = \log$, we discovered that the KL-divergence is optimal also within Bregman divergences. On the other hand, we found out that significant improvements in performance come from modifying $g$ rather than the KL-divergence. We have observed similar results in preliminary MuJoCo experiments where we replaced the Bregman divergence with an $f$-divergence, that is that the KL-divergence is optimal among $f$-divergences.

## F Further discussion of $\omega$-potentials

We show here two examples of Bregman divergence induced by an $\omega$-potential mirror map, that is when $\phi(x) = e^{x-1}$ and when $\phi(x) = x$. If $\phi(x) = e^{x-1}$, the associated mirror map is defined as

$$h_\phi(\pi(\cdot|s)) = \sum_{a \in \mathcal{A}} \int_1^{\pi(a|s)} \phi^{-1}(x)dx = \sum_{a \in \mathcal{A}} \int_1^{\pi(a|s)} (\log(x) + 1)dx$$
$$= \sum_{a \in \mathcal{A}} \pi(a \mid s) \log(\pi(a \mid s)) - \pi(a \mid s) + \pi(a \mid s)$$
$$= \sum_{a \in \mathcal{A}} \pi(a \mid s) \log(\pi(a \mid s)),$$

which is the negative entropy. Plugging this expression in the definition of Bregman divergence we obtain

$$\mathcal{D}_h(x, y) = h(x) - h(y) - \langle \nabla h(y), x - y \rangle$$
$$= \sum_{a \in \mathcal{A}} x_a \log(x_a) - y_a \log(y_a) - (\log(y_a) - y_a)(x_a - y_a)$$
$$= \sum_{a \in \mathcal{A}} x_a \log(x_a/y_a),$$

which is the definition of the KL-divergence. If $\phi(x) = 2x$, the associated mirror map is defined as

$$h_\phi(\pi(\cdot|s)) = \sum_{a \in \mathcal{A}} \int_1^{\pi(a|s)} \phi^{-1}(x)dx = \sum_{a \in \mathcal{A}} \int_1^{\pi(a|s)} 2x dx = \sum_{a \in \mathcal{A}} \pi(a \mid s)^2,$$

which is the $\ell_2$-norm. Plugging this expression in the definition of Bregman divergence we obtain

$$\mathcal{D}_h(x,y) = h(x) - h(y) - \langle \nabla h(y), x - y \rangle = \sum_{a \in \mathcal{A}} x_a^2 - y_a^2 - (2y_a)(x_a - y_a) = \sum_{a \in \mathcal{A}} (x_a - y_a)^2,$$

which is the definition of the Euclidean distance.

# G Further discussion on Evolution Strategies

Evolution Strategies (ES) represent a powerful, backpropagation-free method for optimizing complex functions, that has been particularly successful in the context of long-horizon, noisy, and bi-level optimization tasks such as RL and meta-RL. ES, and in particular the OpenAI-ES algorithm (Salimans et al., 2017), rely on perturbation-based sampling to estimate gradients without requiring backpropagation through the entire computational graph. This feature makes ES well-suited for tasks with long computational graphs, for instance algorithms with many updates, where, due to memory constraints, traditional gradient-based methods have to resort to gradient truncation, introducing bias (Werbos, 1990; Metz et al., 2022; Liu et al., 2022).

In our setting, we use ES to search for the best $\psi$ and $\phi^{-1}$ within the parametrized class introduced in Section 3.1, so that an agent trained using the objective in (11) achieves the highest value. Denote by $\zeta$ the parameters of $\psi$ and $\phi^{-1}$ and by $\pi^\zeta$ the final policy obtained optimizing the objective in (11) when using the parametrized $\psi$ and $\phi^{-1}$. Lastly, let $F(\zeta)$ be the expected cumulative reward (or value) of $\pi^\zeta$, i.e. $F(\zeta) = \mathbb{E}_{\tau \sim (\mu, \pi^\zeta, P)} r(\tau)$. At each iteration, we estimate the gradient $\nabla_\zeta F(\zeta)$ as

$$\mathbb{E}_{\epsilon \sim \mathcal{N}(0, I_d)} \left[ \frac{\epsilon}{2\sigma} (\widehat{F}(\zeta + \sigma\epsilon) - \widehat{F}(\zeta - \sigma\epsilon)) \right], \tag{19}$$

where $\mathcal{N}(0, I_d)$ is the multivariate normal distribution, $d$ is the number of parameters, $\widehat{F}$ is an estimate of $F$, and $\sigma > 0$ is a hyperparameter regulating the variance of the perturbations.

# H Further discussion on Online vs Offline Methods

In the domain of RL and preference optimization, the choice between online and offline algorithms presents a critical trade-off, influencing computational efficiency, data requirements, and generalization capabilities. Online methods, such as PPO, iteratively collect and incorporate new data during training. These inherently support exploration of the environment, enabling the discovery of novel strategies or behaviors that are not captured in pre-existing datasets. However, they need feedback for each generated "trajectory" (or response, in the LLM case), which might be expensive to obtain. Online methods are also more complex and particularly sensitive to hyperparameters, often requiring meticulous tuning for stability and efficiency.

Offline algorithms, such as DPO and its variants, rely entirely on pre-collected datasets. These methods are designed for efficiency and simplicity: they don't require any additional feedback from users and are therefore particularly effective in scenarios where feedback is delayed or unavailable. However, the reliance on static datasets means offline methods may struggle to generalize beyond the training data, particularly if the distribution shift between the training dataset and test time distribution is significant. Additionally, the performance of the algorithm is closely tied to the quality of the training dataset: noisy, biased, or corrupt datasets can severely degrade performance, as these methods cannot mitigate such issues through exploration or resampling.

In summary, RLHF (i.e., online) is considered the superior approach, particularly when substantial amounts of online labels are accessible. This makes it the industry standard (Xu et al., 2024b). While DPO has been theoretically equated to optimizing using PPO and a reward model trained on an offline dataset, recent empirical research (Tang et al., 2024) has challenged this notion. These studies have demonstrated that online methods, such as PPO, consistently outperform offline methods like DPO. This superiority is attributed to the benefits of on-policy sampling.

While DPO has occasionally outperformed PPO, it's important to note that several studies (Xu et al., 2024b; Song et al., 2024) have consistently shown PPO's overall superiority. DPO's relative strength lies in its simpler training regime, which avoids the complexities associated with reward model inaccuracies. However, DPO's performance is significantly limited by its sensitivity to distribution

Table 5: **Three Legged Ant (TLA)**. Performance of existing and discovered MPO algorithms on TLA. For each algorithm-dataset combination, we report the average value and standard error of 25 trained agents. For each discovered MPO algorithm, we specify on which setting it was discovered and report its performance across all settings (with fixed hyperparameters). We report in bold the highest (or two highest if their confidence interval overlaps) average performance, for each setting.

| | Base | Noisy ($\varepsilon = 0.1$) | Noisy ($\varepsilon = 0.2$) | Noisy ($\varepsilon = 0.3$) |
|---|---|---|---|---|
| RRHF | $2789 \pm 285$ | $1730 \pm 442$ | $749 \pm 498$ | $330 \pm 552$ |
| SLiC-HF | $3255 \pm 66$ | $2329 \pm 289$ | $1964 \pm 116$ | $1135 \pm 224$ |
| DPO | $3528 \pm 58$ | $3082 \pm 80$ | $2530 \pm 97$ | $1519 \pm 140$ |
| IPO | $3618 \pm 44$ | $3162 \pm 66$ | $2392 \pm 136$ | $1133 \pm 115$ |
| CPO | $3450 \pm 55$ | $2967 \pm 58$ | $2427 \pm 44$ | $2000 \pm 35$ |
| ORPO | $3087 \pm 322$ | $2841 \pm 50$ | $2359 \pm 43$ | $1953 \pm 37$ |
| R-DPO | $2606 \pm 65$ | $2099 \pm 50$ | $1740 \pm 36$ | $1667 \pm 24$ |
| SimPO | $3683 \pm 78$ | $2314 \pm 752$ | $118 \pm 715$ | $-3828 \pm 341$ |
| SFT | $3287 \pm 62$ | $2733 \pm 45$ | $2345 \pm 37$ | $2049 \pm 33$ |
| KTO | $1534 \pm 34$ | $1551 \pm 31$ | $1531 \pm 49$ | $1442 \pm 35$ |
| f-DPO (Jensen-Shannon) | $3621 \pm 50$ | $3192 \pm 76$ | $2494 \pm 101$ | $1633 \pm 123$ |
| *Our algorithms* ↓ | | | | |
| LPO | $3774 \pm 102$ | $3617 \pm 69$ | $2705 \pm 370$ | $1569 \pm 156$ |
| *With $g = \log \sigma$* | | | | |
| 1S-MPO (mixed-quality) | $3206 \pm 330$ | $1319 \pm 714$ | $-1625 \pm 944$ | $-3967 \pm 382$ |
| 1S-MPO (noisy, $\varepsilon = 0.1$) | $3789 \pm 60$ | $\mathbf{3813 \pm 47}$ | $3280 \pm 83$ | $3279 \pm 83$ |
| 2S-MPO (mixed-quality) | $3595 \pm 57$ | $3197 \pm 58$ | $2487 \pm 110$ | $1687 \pm 58$ |
| 2S-MPO (noisy, $\varepsilon = 0.1$) | $3551 \pm 58$ | $3190 \pm 62$ | $2552 \pm 94$ | $1569 \pm 122$ |
| *With parametrized $g$* | | | | |
| 1S-MPO (mixed-quality) | $3560 \pm 333$ | $3371 \pm 410$ | $3230 \pm 259$ | $2681 \pm 251$ |
| 2S-MPO (mixed-quality) | $3736 \pm 51$ | $3488 \pm 73$ | $2992 \pm 87$ | $2253 \pm 125$ |
| 1S-MPO (noisy, $\varepsilon = 0.1$) | $\mathbf{3861 \pm 79}$ | $3724 \pm 59$ | $2845 \pm 365$ | $1771 \pm 107$ |
| 2S-MPO (noisy, $\varepsilon = 0.1$) | $3701 \pm 52$ | $3490 \pm 95$ | $2886 \pm 127$ | $2074 \pm 136$ |
| 1S-MPO (noisy, $\varepsilon = 0.3$) | $\mathbf{3931 \pm 69}$ | $\mathbf{3834 \pm 82}$ | $\mathbf{3735 \pm 84}$ | $\mathbf{3417 \pm 82}$ |

shift, especially when the offline preference data lacks diversity (Song et al., 2024). This limitation becomes particularly evident when querying the model with out-of-distribution data, a common challenge for methods relying solely on offline data. To mitigate this issue, DPO-iter (Xu et al., 2024b), which incorporates online data, has been proposed as a potential solution.

# I   MuJoCo Additional Results

## I.1   TLA

We display additional results for the TLA task in Table 5. With respect to Table 1, we replace the mixed-quality setting with a noisy setting where the noise parameter $\varepsilon$ is set to 0.2. We also include the KTO (Ethayarajh et al., 2024) and $f$-DPO (Jensen-Shannon) (Wang et al., 2023) algorithms.

## I.2   Simplified expression for discovered objectives

Below, we report a simplified version of the objectives discovered for the shuffled and noisy ($\epsilon = 0.3$) settings, when $g$ is parametrized:

$$\mathcal{L}(\tau_w, \tau_l) = 0.82\mathcal{L}_{\text{SFT}}(\tau_w) + 1.7(\log(\pi_\theta(\tau_w)) - \log(\pi_\theta(\tau_l)))$$
$$+ 0.33(\log(\pi_\theta(\tau_w)) - \log(\pi_\theta(\tau_l)))^2 + 0.36(\log(\pi_\theta(\tau_w)) - \log(\pi_\theta(\tau_l)))^3$$
$$\mathcal{L}(\tau_w, \tau_l) = 0.82\mathcal{L}_{\text{SFT}}(\tau_w)$$
$$+ \max(1.39(\log(\pi_\theta(\tau_w)) - \log(\pi_\theta(\tau_l)))^2, 0.12(\log(\pi_\theta(\tau_w)) - \log(\pi_\theta(\tau_l))))$$

Table 6: **Hopper**. Perfomance of existing MPO algorithms on the Hopper setting. The agent is randomly initialised.

|        | Base          | Mixed Quality | Noisy ($\epsilon = 0.1$) |
|--------|---------------|---------------|--------------------------|
| DPO    | $1796 \pm 78$ | $458 \pm 82$  | $693 \pm 131$            |
| IPO    | $2049 \pm 13$ | $1606 \pm 91$ | $739 \pm 118$            |
| CPO    | $2078 \pm 12$ | $1078 \pm 35$ | $1813 \pm 33$            |
| ORPO   | $2022 \pm 15$ | $1039 \pm 20$ | $1710 \pm 33$            |
| SimPO  | $2027 \pm 15$ | $1460 \pm 94$ | $1794 \pm 65$            |

### I.3 Hopper Tasks

We consider a second set of simulations on MuJoCo, based on the Hopper environment. As for the previous sections, our experiments are implemented in JAX (Bradbury et al., 2018) using the `brax` (Freeman et al., 2021) and evosax (Lange, 2022) libraries. We report our hyper-parameters in Appendix K.

Differently from the TLA task, we consider a setting where the agent is randomly initialized and needs to learn a policy from scratch. On `Hopper`, we train an agent with an expected cumulative reward of 2100 (the expert agent) and an agent with an expected cumulative reward of 900 (the bad agent). We generate the preference datasets in the same way we do for TLA, with the exceptions that the number of rows is 5120 and the trajectories are generated by either the expert or the bad agent. We consider the same three variations of the preference datasets used in TLA, where the expert agent corresponds to the target agent, and the bad agent corresponds to the original agent. We include more data compared to the TLA setting as it takes more datapoints for the agent to learn from scratch rather than to adapt to a slightly different objective (like in the TLA case).

Our Hopper results confirm our conclusions in the TLA setting. All algorithms but DPO come close to matching the performance of the expert agent (2100), with CPO being the best. We can see SimPO is the only algorithm that significantly outperforms the bad agent (performance of 900) in the Mixed Quality setting (the $\gamma$ for SimPO was set very high, $\gamma = 10$, as a lower value significantly limited performance).

### I.4 Further MuJoCo Analsys

In addition to the noisy dataset, we also considered a **bad judge** setting, where the judge would be more likely to swap the label of a pair of trajectories if their ground truth rewards were closer to each other. This is practically implemented as an increase in the temperature of the Bradley-Terry judge. However, we did not notice significantly different results compared to the simple noisy setting, therefore detailed results are not reported.

## J  Further discussion on Meta-Learning Algorithms

Meta-learning, or "learning to learn", has been extensively employed to automate the design of algorithms that can either adapt rapidly with minimal data samples or generalize effectively to unseen data, tasks, or environments. The development of broadly applicable algorithms is particularly critical in the context of preference optimization for LLMs. Here, LLMs are fine-tuned on relatively small datasets of offline data but must generalize to a virtually infinite range of potential user queries. Prior work in meta-learning has demonstrated success in developing generalizable optimization algorithms and loss functions (Lu et al., 2022; Jackson et al., 2024; Lu et al., 2024; Goldie et al., 2024; Kirsch et al., 2020).

At its core, meta-learning is defined as a bilevel optimization problem with an inner and an outer loop. The inner loop consists in an iterative optimization algorithm that trains agents to solve a predetermined task given a set of meta-parameters. The outer loop consists in evaluating the agents trained in the inner loop and update the meta-parameters accordingly, following some optimization method like second order gradient descent (Finn et al., 2017). The evaluation of the agents is typically done on a held-out dataset in supervised learning or by sampling trajectories on the environment simulator in RL (Lu et al., 2022; Jackson et al., 2024). In our setting, the inner loop is the offline preference optimization algorithm, while the outer loop is the agent evaluation on the environment (online) and the update of the meta-parameters $\zeta$.

# K MuJoCo Hyper-parameters

We give the hyper-parameters we use for training. The hyper-parameters specific to each algorithm are tuned for each task-data type combination. All the experiments were conducted on 4 NVIDIA L40S GPUs.

Table 7: Hyper-parameter settings for PO.

| Parameter | Value |
|-----------|-------|
| Number of epochs | 12 |
| Minibatch size | 2 |
| Learning rate | 1e-3 |
| Max gradient norm | 1.3 |

Table 8: Hyper-parameter settings of OpenAI-ES.

| Parameter | Hopper | TLA |
|-----------|--------|-----|
| Population Size | 256 | 256 |
| Number of generations | 128 | 256 |
| Sigma init | 0.03 | 0.03 |
| Sigma Decay | 0.999 | 0.999 |
| Learning rate | 0.02 | 0.02 |

# L LLM Hyper-parameters

We give the hyper-parameters we use for LLM training. All the experiments were conducted on 4 NVIDIA L40S GPUs.

Table 9: Hyper-parameter settings for LLM Training.

| Parameter | Value |
|-----------|-------|
| Gradient Accumulation Step | 32 |
| Batch Size | 2 |
| Total Batch Size | 64 |
| LoRA | Yes |
| LoRA Rank | 128 |
| LoRA Alpha | 256 |
| Lora Dropout | 0.05 |
| Max length | 2048 |

