# OpenReview forum: "Meta-Learning Objectives for Preference Optimization"
_NeurIPS.cc/2025/Conference — NeurIPS 2025 poster_

### Official Review · Reviewer_B54D · 2025-06-30

**Clarity:** 3
**Significance:** 2
**Originality:** 3
**Rating:** 4
**Confidence:** 2

**Summary:**

This paper proposes a unified formulation of the existing Preference Optimisation (PO) algorithm. Significantly, it replaces the original KL divergence with the Bregman divergence and introduces an evolutionary term to control the percentage of training progress from SFT to PO.  Based on the unified formulation (meta-learning objective, such as 1step/2-step MPO and Temporally-Aware MPO, TA-MPO), they evaluate existing PO algorithms in a synthetic experiment environment, MuJoCo, in which you can control the reward structure as the reward datasets are well-defined. Beyond the synthetic setting, the authors also show the results when transferring to LLM. Specifically,  they implement the learning objective (TA-MPO) and compare with existing PO algorithms, such as DPO, SimPO, etc. The experimental results show the effectiveness of TA-MPO.

**Questions:**

1. As far as I understand, this meta-objective (I) replaces KL with Bregman divergence (2), introducing an adaptive factor transition from SFT to PO, so what are the separate effects for the two designs?
2. Why TA-MPO (1), TA-MPO (2)  perform very differently? any hypermeter suggestions in different practical settings?

**Ethical Concerns:**

["NO or VERY MINOR ethics concerns only"]

**Final Justification:**

I keep my original positive evaluation score, as it introduces an evolutionary term to control the percentage of training progress from SFT to PO, and empirical results in a controlled environment show its efficiency.

**Limitations:**

Yes

**Quality:**

3

**Strengths And Weaknesses:**

Strengths:
1. This paper proposes a unified theoretical formulation of existing preference optimisation algorithms, which provide a comprehensive and fundamental perspective to compare and improve the PO algorithms.
2. The meta-objective shows effectiveness in both synthetic and LLM PO settings.


Weakness:
1. The verification in LLM settings is rather limited compared with the synthetic settings, so it is unclear if the objective can work well and be generalisable in real LLM training. Also, the performances of TA-MPO (1), TA-MPO (2) vary a lot.

---

> ### Author Rebuttal · Authors · 2025-07-30
>
> We thank the reviewer for their valuable feedback. Please find a reply to your concerns below.
>
> **[Separate effects of Bregman divergences and temporal awareness]** The objective in Equation (12) that we propose for LLM alignment takes advantage of the insights we developed on MuJoCo. The first is that for $g=\mathrm{log}$ we rediscover the KL-divergence, so we set $\phi^{-1}(x)= \mathrm{log}(x)$, as in DPO or in SimPO. The second is that it is important to keep optimizing even when $\pi(\tau_w)>\pi(\tau_l)$, which is why set $g$ so that it introduces a constant $\gamma$ as in SimPO. The third is that objectives with an SFT term are more robust to noise, which is why we add an SFT term in Equation (12). The fourth is that a gradual transition from SFT to PO helps performance. In the next paragraph, we show that each of the last three insights leads to improved performance in the LLM setting.
>
> **[Additional LLM experiments]** We have performed additional LLM experiments to better understand the influence of different components of TA-MPO. In particular, we have tested SimPO with $\gamma = 0$, SimPO with $\gamma>0$, and SFT + SimPO with $\gamma>0$, where the last one consists of a one-step objective made of the addition between the SFT and the SimPO loss. The settings and hyperparameters are the same as in Section 5 of our paper.  We use the gemma-7b model and show the win-rate as main metric and the length controlled win-rate in parenthesis.  We report the results below.
> | Algorithm   | $\gamma$  | dpo-mix-7k       | capybara-7k     |
> |-------------|----|------------------|------------------|
> | SimPO       | 0  | 30.50 (26.06)    | 27.52 (25.12)    |
> | SimPO       | 1  | 30.62 (25.35)    | 27.52 (25.54)    |
> | SimPO       | 2  | 32.50 (25.80)    | 27.95 (25.71)    |
> | SimPO       | 5  | 33.42 (28.22)    | 27.02 (25.07)    |
> | SimPO       | 10 | 33.60 (27.12)    | 27.39 (25.00)    |
> | SFT+SimPO   | 1  | 30.31 (26.55)    | 35.07 (28.25)    |
>
> These new results show that having a large $\gamma$, which encourages large updates even when $\pi(\tau_w)>\pi(\tau_l)$, or adding an SFT term to the PO loss are both helpful in improving performance. The performance we report for TA-MPO in our paper is still the highest, meaning that temporal awareness also contributes to a better performance. We can therefore confirm that each of the insights we gained in the MuJoCo benchmark transfers to the LLM alignment setting.
>
>
>
> **[Performance of different TA-MPO algorithms]** The difference in performance of the different TA-MPO algorithms is due to the fact that they induce different learning rate schedules for the SFT and PO terms. As shown by [1], these types of hyperparameters can have a strong influence on the final performance of the algorithm, which is why we explored a few variations for the temporal awareness schedule $\alpha$.
>
> [1] Meng, Yu, Mengzhou Xia, and Danqi Chen. "Simpo: Simple preference optimization with a reference-free reward." Neural Information Processing Systems, 2024.

---

### Official Review · Reviewer_cgc5 · 2025-07-01

**Clarity:** 3
**Significance:** 3
**Originality:** 3
**Rating:** 5
**Confidence:** 2

**Summary:**

In this paper, the authors introduce a generalized class of direct preference alignment approaches that build on some theoretical techniques from information geometry. Specifically, in contrast to classical RLHF objective (Eq. 4), they replace the KL-divergence term with a Bregman divergence term (Eq. 7) based on "mirror maps" (the details of which are included in 2.2. While I wasn't familiar with these concepts, I found this part easy to follow). They key advantage of this formulation over a standard KL-divergence term is summarized starting on line 140: "When the Bregman divergence is employed as a regularization term in optimization problems, tuning the mirror map allows us to control the geometry of the updated of the parameters to be optimized, determining when to take large or small updates..."

Most interestingly, many mirror maps can be defined that induce different Bregman divergences, each of which will enforce different types of regularization (this includes a temporal-aware variant, the learns to switch between different losses). They show that DPO and ORPO, two popular direct preference alignment approaches, employ a particular set of mirror maps (their full objective and derivation of DPO and ORPO importantly relies on Theorem 3, which follows a similar derivation to what's used in the original DPO). What's more, they propose learning new mirror maps using meta-learning via the OpenAI-ES method in order to discovery preference-tailored losses.

Their experiment setting is non-standard, and first involves what they call "MuJoCO experiments" and variations of the ant simulation (clarifying what MuJoCO is and stands for early on would be helpful for those who are not familiar). In brief summary, they use the MuJoCO environment to generate different ant simulation scenarios with specific properties (e.g., stylistic variations, systematically injected noise) and synthetically generate preference data from these scenarios. The intuition (as I understand) is use such data to demonstrate how their approach is able to adapt to the pecularities of each preference distribution and learn the corresponding mirror maps via OpenAI-ES. They find that their approach is able to effectively discover/meta-learn improved losses (Table 1). To show that their approach also works in a more conventional LLM preference optimization setting, they report small scale experiments in Table 2 where their approach is competitive with a comprehensive set of standard approaches (CPO, ORPO, DPO, SimPO).

**Questions:**

-- Why are the length controlled win-rates put in the parethenses in Table 2, shouldn't this be the main metric to report? Please clarify.

-- Can you explain why you only use these 7k preference mixtures? Do you think this would be the same if you scaled this? Have you tried any scaled experiments?

**Ethical Concerns:**

["NO or VERY MINOR ethics concerns only"]

**Final Justification:**

I found that the rebuttal clarified the important questions I had. Given that my score was already high, I decided to keep it that way, also after considering the feedback of other reviewers and their rebuttals.

**Limitations:**

yes

**Quality:**

3

**Strengths And Weaknesses:**

# strengths

-- A new theoretical framework for direct preference optimization from information geometry (MPO) that broadens our understanding and the scope of current approaches. This approach not only provides an original view on existing losses, but provides a means for defining many new classes of losses.

-- An empirical demonstration that such a framework can yield improved losses, especially through meta- and evolutionary learning. I could imagine that this part about learning the parameters of their MPO could influence many follow up studies, especially following up on their small scale "LLM transfer" study in Section 5.

# weaknesses

-- (minor) Given the non-standard nature of their MuJoCO expeirments, a figure illustrating their data generation process and a running example would be very helpful, since the details are otherwise hard to follow.

-- (minor) I also found figure 5 to be confusing, an intuition for why the absolute value of the gradient is meaninful to show seems to be missing. Please clarify this.

---

> ### Author Rebuttal · Authors · 2025-07-30
>
> We thank the reviewer for their valuable feedback. Please find a reply to your concerns below.
>
> **[Objective of Figure 5]** Understanding the gradient of an objective function is fundamental when analyzing gradient-descent-based methods. We plot the gradient of the objective functions to provide a visual representation of their behavior and to understand where they induce large or small updates. We plot the absolute value as the gradient w.r.t. the chosen trajectory is always positive and the gradient w.r.t. the rejected trajectory is always negative.
>
> **[LLM benchmark metrics]** We agree with the reviewer, we will report the length controlled win rates as the main metric and report the win rates in parentheses.
>
> **[LLM benchmark]** We performed our experiments on datasets with 7k samples as our computational resources do not allow for a comprehensive comparison on a larger dataset like ultrafeedback [1]. We highlight that other works have performed their LLM experiments exclusively on small datasets (e.g. [2]) and that [3] have shown that training on a smaller but higher quality dataset can lead to a superior performance than training on a larger dataset.
>
> **[Data generation figure]** Unfortunately, this year's guidelines prevent us from sharing a figure in the rebuttal. We will include a figure explaining the data generation procedure in a camera ready version of this paper.
>
> **[Additional LLM experiments]** We have performed additional LLM experiments to better understand the influence of different components of TA-MPO. In particular, we have tested SimPO with $\gamma = 0$, SimPO with $\gamma>0$, and SFT + SimPO with $\gamma>0$, where the last one consists of a one-step objective made of the addition between the SFT and the SimPO loss. The settings and hyperparameters are the same as in Section 5 of our paper. We use the gemma-7b model and show the win-rate as main metric and the length controlled win-rate in parenthesis, for consistency. We report the results below.
> | Algorithm   | $\gamma$  | dpo-mix-7k       | capybara-7k     |
> |-------------|----|------------------|------------------|
> | SimPO       | 0  | 30.50 (26.06)    | 27.52 (25.12)    |
> | SimPO       | 1  | 30.62 (25.35)    | 27.52 (25.54)    |
> | SimPO       | 2  | 32.50 (25.80)    | 27.95 (25.71)    |
> | SimPO       | 5  | 33.42 (28.22)    | 27.02 (25.07)    |
> | SimPO       | 10 | 33.60 (27.12)    | 27.39 (25.00)    |
> | SFT+SimPO   | 1  | 30.31 (26.55)    | 35.07 (28.25)    |
>
> These new results show that having a large $\gamma$, which encourages large updates even when $\pi(\tau_w)>\pi(\tau_l)$, or adding an SFT term to the PO loss are both helpful in improving performance. The performance we report for TA-MPO in our paper is still the highest, meaning that temporal awareness also contributes to a better performance. We can therefore confirm that each of the insights we gained in the MuJoCo benchmark transfers to the LLM alignment setting.
>
> [1] HuggingFaceH4/ultrafeedback\_binarized
>
> [2] Lu, Chris, et al. "Discovering preference optimization algorithms with and for large language models." Neural Information Processing Systems, 2024.
>
> [3] Deng, Xun, et al. "Less is more: Improving llm alignment via preference data selection." arXiv preprint arXiv:2502.14560, 2025.

---

> > ### Comment · Reviewer_cgc5 · 2025-08-06
> >
> > Thank you for these additional details, they do address my questions and concerns is a satisfactory manner. I will keep my positive score.

---

### Official Review · Reviewer_FmSK · 2025-07-03

**Clarity:** 3
**Significance:** 3
**Originality:** 2
**Rating:** 4
**Confidence:** 4

**Summary:**

This paper studies preference optimization (PO) algorithms for aligning both reinforcement learning agents in simulated control environments and large language models (LLMs) using human feedback. Motivated by the high cost and instability of evaluating PO algorithms directly on LLMs, the authors design a MuJoCo-based diagnostic benchmark, simulating preference datasets with varying quality and noise. The authors empirically show that their discovered objectives outperform baselines on MuJoCo TLA tasks, particularly in noisy and mixed-quality datasets, and that a variant (TA-MPO) yields competitive or superior results in LLM alignment.

**Questions:**

Please refer to Strengths And Weaknesses

**Ethical Concerns:**

["NO or VERY MINOR ethics concerns only"]

**Final Justification:**

Thank authors for the response, I will maintain my positive score. Moreover, my current assessment is that this is a borderline paper.

**Limitations:**

Yes

**Quality:**

3

**Strengths And Weaknesses:**

Strengths

1. The authors alleviate an important evaluation challenge in PO by introducing a controlled and interpretable MuJoCo benchmark to probe PO algorithms’ behaviors under varying data noise and quality, enabling better comparisons than LLMs alone.

2. The MPO framework is interesting, extending and generalizing well-known PO algorithms (DPO, ORPO) via mirror maps and Bregman divergences.

3. The approach shows robustness and improved performance in high-noise and mixed-quality data cases.

4. This paper provides gradients for SimPO, ORPO, and discovered MPO losses, illustrating where these losses induce significant parameter updates; such visual inspection aids interpretability of the algorithms’ behavior.


Weaknesses

1. While the paper claims practical transferability to LLM alignment, the experiments in Table 2 only cover AlpacaEval. The degree of improvement, although sometimes statistically significant, is not overwhelming. It’s not clear how robust TA-MPO will be across other important alignment tasks, adversarial data, or human preferences beyond the datasets tested.

2. As noted in Section 5, algorithms discovered with parameterized $g$ overfit the MuJoCo setting and transfer poorly to LLMs, raising concerns about generalizability and task-specific tuning rather than finding universal improvements.

3. Table 2’s reliance on an automatic judge (Llama-3.1-70B-Instruct) rather than diverse human annotators or more comprehensive human-based preference collection somewhat weakens claims of improved LLM alignment in practice.

---

> ### Author Rebuttal · Authors · 2025-07-30
>
> We thank the reviewer for their valuable feedback. Please find a reply to your concerns below.
>
> **[LLM benchmarks]** We remark that our LLM experiments have been performed in three different settings, for different combinations of language model (gemma-7b and mistral-7b) and preference dataset (dpo-mix-7k and capybara-7k). Only the evaluation was done on AlpacaEval, which is a standard benchmark for evaluating LLM performance (see for instance [1, 2]). We also highlight that automatic judges present a high level of agreement with human judges, as reported in the AlpacaEval repository.
>
> **[Transfer to LLM alignment of discovered objective]** While the meta-learned optimizers can overfit to particular tasks, our paper shows that we can still extract generalizable knowledge from them. We provide additional results regarding the transfer of the insights obtained from MuJoCo below.
>
> **[Additional LLM experiments]** We have performed additional LLM experiments to better understand the influence of different components of TA-MPO. In particular, we have tested SimPO with $\gamma = 0$, SimPO with $\gamma>0$, and SFT + SimPO with $\gamma>0$, where the last one consists of a one-step objective made of the addition between the SFT and the SimPO loss. The settings and hyperparameters are the same as in Section 5 of our paper.  We use the gemma-7b model and show the win-rate as main metric and the length controlled win-rate in parenthesis. We report the results below.
> | Algorithm   | $\gamma$  | dpo-mix-7k       | capybara-7k     |
> |-------------|----|------------------|------------------|
> | SimPO       | 0  | 30.50 (26.06)    | 27.52 (25.12)    |
> | SimPO       | 1  | 30.62 (25.35)    | 27.52 (25.54)    |
> | SimPO       | 2  | 32.50 (25.80)    | 27.95 (25.71)    |
> | SimPO       | 5  | 33.42 (28.22)    | 27.02 (25.07)    |
> | SimPO       | 10 | 33.60 (27.12)    | 27.39 (25.00)    |
> | SFT+SimPO   | 1  | 30.31 (26.55)    | 35.07 (28.25)    |
>
> These new results show that having a large $\gamma$, which encourages large updates even when $\pi(\tau_w)>\pi(\tau_l)$, or adding an SFT term to the PO loss are both helpful in improving performance. The performance we report for TA-MPO in our paper is still the highest, meaning that temporal awareness also contributes to a better performance. We can therefore confirm that each of the insights we gained in the MuJoCo benchmark transfers to the LLM alignment setting.
>
> [1] Lu, Chris, et al. "Discovering preference optimization algorithms with and for large language models." Neural Information Processing Systems, 2024.
>
> [2] Meng, Yu, Mengzhou Xia, and Danqi Chen. "Simpo: Simple preference optimization with a reference-free reward." Neural Information Processing Systems, 2024.

---

> > ### Comment · Reviewer_FmSK · 2025-08-08
> >
> > Thank you for the response. I will maintain my positive score.

---

### Official Review · Reviewer_wh5w · 2025-07-03

**Clarity:** 2
**Significance:** 2
**Originality:** 3
**Rating:** 4
**Confidence:** 3

**Summary:**

This paper proposes a new framework, Mirror Preference Optimization (MPO), for aligning models with human preferences. Instead of evaluating Preference Optimization (PO) methods directly on large language models (LLMs), which is costly and noisy, the authors construct a controlled benchmark using MuJoCo environments to simulate preference datasets. They formalize a general class of PO algorithms using mirror descent and employ evolutionary strategies to discover algorithms tailored to various dataset properties (e.g., noise, quality). These discovered algorithms outperform existing PO methods in MuJoCo settings. Leveraging insights from these experiments, they propose a temporally-aware MPO (TA-MPO) algorithm, which shows improved performance in real LLM alignment tasks.

**Questions:**

1. Can you explain the insights that allow transfer from MuJoCo to LLMs?
2. Can you do more LLM experiments?
3. Why do you use Bregman divergence instead of f-divergence?

**Ethical Concerns:**

["NO or VERY MINOR ethics concerns only"]

**Final Justification:**

While initially I have some concerns, they are addressed by the authors in the discussion period. Thus, I recommend acceptance.

**Limitations:**

Yes.

**Paper Formatting Concerns:**

No.

**Quality:**

2

**Strengths And Weaknesses:**

**Strengths**

Proposes a unified framework (MPO) that generalizes DPO/ORPO and allows algorithm discovery via mirror descent and ES.

Experiments on MuJoCo are systematic and cover different noise and data quality settings.

The discovered algorithms outperform baselines in these synthetic benchmarks.

**Weaknesses**

Related work is incomplete—missing citations like SPO and several recent LLM-specific PO methods.

Evaluating PO methods on simple benchmarks is not novel and has been done before.

The claimed transfer from MuJoCo to LLM lacks clear insight or analysis showing what actually generalizes.

LLM experiments are limited and only form a small part of the paper.

---

> ### Author Rebuttal · Authors · 2025-07-30
>
> We thank the reviewer for their valuable feedback. Please find a reply to your concerns below.
>
> **[Additional LLM experiments]** We have performed additional LLM experiments to better understand the influence of different components of TA-MPO. In particular, we have tested SimPO with $\gamma = 0$, SimPO with $\gamma>0$, and SFT + SimPO with $\gamma>0$, where the last one consists of a one-step objective made of the addition between the SFT and the SimPO loss. The settings and hyperparameters are the same as in Section 5 of our paper. We use the gemma-7b model and show the win-rate as main metric and the length controlled win-rate in parenthesis. We report the results below.
> | Algorithm   | $\gamma$  | dpo-mix-7k       | capybara-7k     |
> |-------------|----|------------------|------------------|
> | SimPO       | 0  | 30.50 (26.06)    | 27.52 (25.12)    |
> | SimPO       | 1  | 30.62 (25.35)    | 27.52 (25.54)    |
> | SimPO       | 2  | 32.50 (25.80)    | 27.95 (25.71)    |
> | SimPO       | 5  | 33.42 (28.22)    | 27.02 (25.07)    |
> | SimPO       | 10 | 33.60 (27.12)    | 27.39 (25.00)    |
> | SFT+SimPO   | 1  | 30.31 (26.55)    | 35.07 (28.25)    |
>
> These new results show that having a large $\gamma$, which encourages large updates even when $\pi(\tau_w)>\pi(\tau_l)$, or adding an SFT term to the PO loss are both helpful in improving performance. The performance we report for TA-MPO in our paper is still the highest, meaning that temporal awareness also contributes to a better performance. We can therefore confirm that each of the insights we gained in the MuJoCo benchmark transfers to the LLM alignment setting.
>
> **[Related works]** We provide discussions on related works in the Introduction, in Section 2.1, and in Appendices A and B. Additionally, we test eight existing preference optimization algorithms on our MuJoCo benchmark. Should the reviewer want us to test more algorithms, we would be glad to test any additional proposed algorithm. As to the proposed SPO method, we found four possible candidates in the literature, which we discuss below. Please let us know which one you would like us to include in our work.
>
> - Self-Play Preference Optimization [1]: the authors propose a method to tackle non-Markovian, intransitive, and stochastic preference models. While their method can be applied to the offline setting, i.e. ours, the authors do not perform any experiment in this setting. Additionally, their method involves estimating a preference function, which does not fall in the DPO-style losses we consider in our work.
>
> - Soft Preference Optimization [2]: the authors propose a DPO-style loss where the KL-divergence penalty between the current and the reference policy does not depend on the preference dataset. The KL-divergence is instead estimated by sampling trajectories from the current policy every $T$ policy updates, for a hyperparameter $T$. While the authors have not released code for their implementation, we can implement this algorithm should it be the one meant by the reviewer.
>
> - Self-supervised Preference Optimization [3]: the authors propose a method to generate synthetic data and to use it alongside the original preference dataset to train an LLM. Although their proposed loss is similar to the DPO loss, methods that involve synthetic data are beyond the scope of our work.
>
> - Step-by-step Preference Optimization [4]: the authors propose a method for post-training diffusion models using preference optimization. The method is tailored to the setting of diffusion models and cannot be straightforwardly applied to our setting.
>
> **[Objective of our MuJoCo benchmark]** Evaluating PO algorithms on simpler environment has been done before in the literature [1, 5]. However, we are not aware of a recent work that has performed a comprehensive comparison of PO algorithms in a simple benchmark, or that has used the simple benchmark to obtain insights transferrable to the LLM alignment setting.
>
>
> **[Bregman divergences vs $f$-divergences]** In our work, we choose to use Bregman divergences because $f$-divergences have already been explored by previous works in the literature [6], while the application of Bregman divergences to PO was missing from the literature.
>
>
>
> [1] Swamy, Gokul, et al. "A Minimaximalist Approach to Reinforcement Learning from Human Feedback." International Conference on Machine Learning, 2024.
>
> [2] Sharifnassab, Arsalan, et al. "Soft preference optimization: Aligning language models to expert distributions." arXiv preprint arXiv:2405.00747, 2024.
>
> [3] Li, Jian, et al. "Self-supervised Preference Optimization: Enhance Your Language Model with Preference Degree Awareness." Findings of the Association for Computational Linguistics: EMNLP 2024.
>
> [4] Liang, Zhanhao, et al. "Aesthetic post-training diffusion models from generic preferences with step-by-step preference optimization." Proceedings of the Computer Vision and Pattern Recognition Conference, 2025.
>
> [5] Xu, Shusheng, et al. "Is dpo superior to ppo for llm alignment? a comprehensive study." arXiv preprint arXiv:2404.10719, 2024.
>
> [6] Wang, Chaoqi, et al. "Beyond reverse kl: Generalizing direct preference optimization with diverse divergence constraints". International Conference on Learning Representations, 2023.

---

> > ### Comment · Reviewer_wh5w · 2025-08-06
> > **Thanks for the rebuttal and additional concerns**
> >
> > I would like to thank the authors for their rebuttal. However, I still have several remaining concerns.
> >
> > First, when I asked “Why do you use Bregman divergence instead of f-divergence?”, I was already aware that f-DPO explores the use of f-divergences. My question was intended to understand the authors’ rationale for exploring Bregman divergences specifically. The justification that “it has not been explored yet” seems insufficient on its own.
> >
> > Second, could the authors clarify why LLaMA-3.1-70B-Instruct was chosen as the judge? What are the implications of this choice on the evaluation?
> >
> > Finally, is it feasible to directly discover algorithms in the LLM settings?

---

> > > ### Author Response · Authors · 2025-08-06
> > >
> > > **[Bregman divergences vs $f$-divergences]** We will provide a more in depth discussion on the difference between Bregman divergences and f-divergences in the setting of preference optimization. Firstly, we note that, if the reference policy is set to be the uniform distribution, Bregman divergences and f-divergences generate equivalent families of algorithms. Let $\phi$ be a 0-potential. In the case of Bregman divergences, we have
> > > $$r(\tau_w)-r(\tau_l)= \phi^{-1}(\pi^\star(\tau_w))-\phi^{-1}(\pi_\mathrm{ref}(\tau_w))-\phi^{-1}(\pi^\star(\tau_l))+\phi^{-1}(\pi_\mathrm{ref}(\tau_l)).$$
> > > If the reference policy is uniform, we obtain
> > > $$r(\tau_w)-r(\tau_l)= \phi^{-1}(\pi^\star(\tau_w))-\phi^{-1}(\pi^\star(\tau_l)).$$
> > > In the case of f-divergences, we have
> > > $$r(\tau_w)-r(\tau_l)= \phi^{-1}(\pi^\star(\tau_w)/\pi_\mathrm{ref}(\tau_w))-\phi^{-1}(\pi^\star(\tau_l)/\pi_\mathrm{ref}(\tau_l)).$$
> > > If the reference policy is uniform, we obtain $$r(\tau_w)-r(\tau_l)= \phi^{-1}(|\mathcal{T}|\pi^\star(\tau_w))-\phi^{-1}(|\mathcal{T}|\pi^\star(\tau_l))$$
> > > $$= \bar{\phi}^{-1}(\pi^\star(\tau_w))-\bar{\phi}^{-1}(\pi^\star(\tau_l)),$$
> > > where $\mathcal{T}$ is the set of all trajectories and $\bar{\phi}^{-1}(x)=\phi^{-1}(|\mathcal{T}|x)$. The two resulting expressions are equivalent, meaning that our 1S-MPO class also includes the case of $f$-divergences. Our experiment on the 1S-MPO class therefore explore both Bregman divergences and f-divergences with ES and Table 1 reports the performance of the best divergence found within both divergence classes.
> > >
> > > If the reference policy is not uniform, then the two algorithmic classes have the KL divergence as the only intersection. In this setting, we have chosen to focus on Bregman divergences as [6] have already shown that, among the $f$-divergences they considered, the KL-divergence typically offers superior alignment performance. Since we wanted to explore a class of algorithms with the objective of finding better alternatives to the KL-divergence, we decided to consider a different generalization of the KL-divergence, i.e. Bregman divergences. In our experiments with $g=\mathrm{log}$, we discovered that the KL-divergence is optimal also within Bregman divergences. On the other hand, we found out that significant improvements in performance come from modifying $g$ rather than the KL-divergence. We will include this discussion in the paper.
> > >
> > > To confirm that the KL-divergence is optimal within f-divergences, we are currently performing additional experiments for 2S-MPO where we set $g=\mathrm{log}$ and replace the Bregman divergence with an f-divergence. That is, we consider the objective
> > > $$ \mathbb{E}\_\mathcal{D}\\;\mathrm{log}\sigma(\phi^{-1}(\pi^\star(\tau_w)/\pi\_\mathrm{ref}(\tau_w))-\phi^{-1}(\pi^\star(\tau_l)/\pi_\mathrm{ref}(\tau_l)))$$
> > > and optimize the function $\phi^{-1}$ using ES. We will include these experiments in the rebuttal if they finish in time.
> > >
> > > **[LLaMa-70B as a judge]** Our choice of using LLaMA-3.1-70B-Instruct as evaluator on AlpacaEval is the result of a cost-benefit analysis. We considered the prohibitive costs associated with using proprietary models like GPT-4 for the extensive evaluation required in our study and the fact that LLaMA-3.1-70B-Instruct is a high-performing an evaluator as well as an open-weight model that can be run locally for a moderate price. The high performance of LLaMA-3.1-70B-Instruct as an evaluator, and in particular its correlation with GPT-4's judgments and human preferences, has been attested in the literature [1, 2], in the AlpacaEval repository, and in blogposts [3, 4]. This strong alignment ensures that our evaluation results, while not based on a proprietary model, are reliable and consistent with established benchmarks.
> > >
> > >
> > > [1] B. Moniri et al. Evaluating the Performance of Large Language Models via Debates. NAACL, 2025.
> > >
> > > [2] P. Wang et al. "Direct judgement preference optimization." arXiv preprint arXiv:2409.14664, 2024.
> > >
> > > [3] R. Raju. Replacing the Judge: Can Llama 405B Outperform GPT4 in the Court of AI?
> > >
> > > [4] Symflower. Is Llama-3 better than GPT-4 for generating tests? And other deep dives of the DevQualityEval v0.4.0.

---

> > > > ### Author Response · Authors · 2025-08-06
> > > >
> > > > **[ES on LLM alignment]** We agree that directly optimizing the loss on an LLM alignment task would be a valuable experiment, but it has high computational costs. Assume we have access to a server with 8 A100 GPUs, where each GPU is capable of training 16 0.5B model (such as Qwen) on dpo-mix-7k in around 20 minutes and evaluating them in about 1 hour. Given a population size of 128 and a number of ES step of 100-200, i.e. the number of steps it took most of our experiments to converge, it would take around 5-10 days to complete an ES run. Unfortunately, due to limited access to the necessary computational resources, we were unable to conduct this experiment prior to the submission deadline. However, we have now initiated this experiment, and while our current hardware limitations mean that it will take approximately two weeks to complete, we are committed to including the results and a thorough analysis in the camera-ready version of our paper. We believe these findings will provide a more comprehensive validation of our work. We thank the reviewer for their valuable suggestion.
> > > >
> > > > Please let us know if there is any remaining concern.

---

> > > > > ### Comment · Reviewer_wh5w · 2025-08-07
> > > > > **Thanks for the rebuttal**
> > > > >
> > > > > I would like to thank the authors for the rebuttal. As they adequately address my concerns, I raise my score to 4.

---

> > > > > > ### Author Response · Authors · 2025-08-09
> > > > > >
> > > > > > We would like to thank the reviewer for increasing their score.
> > > > > >
> > > > > > To follow up on our previous comment regarding the $f$-divergence class, we will clarify in the camera-ready version of the paper that the 1S-MPO class includes the case of $f$-divergences and that the difference between Bregman divergences and $f$-divergences is impactful only within the 2S-MPO class. We have also initiated ES runs for 2S-MPO with $g=\mathrm{log}$ and with parametrized $g$, for all dataset settings, which we will include in the camera-ready version of this paper.

---

> ### Author Response · Authors · 2025-08-05
>
> As the reviewer-author discussion period is coming to an end, we would greatly appreciate it if you could kindly let us know if our rebuttal has resolved your concerns, and if you have any additional questions about our work.

---

### Decision · Program_Chairs · 2025-09-17

**Decision:**

Accept (poster)

**Comment:**

All reviewers agreed on the primary strength of this paper: it introduces a generalisation of existing preference optimisation algorithms that includes DPO, ORPO, and others; and, on a set of controlled MuJoCo experiments demonstrates its efficiency. The meta-learning aspect appears to be able to discover novel approaches that are useful particularly in high-noise settings, and provide some insight into performance of existing hand-designed algorithms. The primary concerns from reviewers were that the transferability to LLMs (or indeed, general applicability to LLMs where these algorithms are usually applied).

However, after the rebuttal phase all reviewers agreed that the paper deserved acceptance (though the scores are still borderline), on the assumption that the final version wil incorporate the additional results and discussion promised by the authors.